# An open resource combining multi-contrast MRI and microscopy in the macaque brain

Amy F. D. Howard ®[1] ✉, Istvan N. Huszar[1], Adele Smart ®[1,2], Michiel Cottaar ®[1], Greg Daubney[3], Taylor Hanayik[1], Alexandre A. Khrapitchev ®[4], Rogier B. Mars ®[1,5], Jeroen Mollink[1], Connor Scott ®[2], Nicola R. Sibson ®[4], Jerome Sallet[3], Saad Jbabdi[1,6] & Karla L. Miller ®[1,6]

Understanding brain structure and function often requires combining data across different modalities and scales to link microscale cellular structures to macroscale features of whole brain organisation. Here we introduce the Big-Mac dataset, a resource combining in vivo MRI, extensive postmortem MRI and multi-contrast microscopy for multimodal characterisation of a single whole macaque brain. The data spans modalities (MRI and microscopy), tissue states (in vivo and postmortem), and four orders of spatial magnitude, from microscopy images with micrometre or sub-micrometre resolution, to MRI signals on the order of millimetres. Crucially, the MRI and microscopy images are carefully co-registered together to facilitate quantitative multimodal analyses. Here we detail the acquisition, curation, and first release of the data, that together make BigMac a unique, openly-disseminated resource available to researchers worldwide. Further, we demonstrate example analyses and opportunities afforded by the data, including improvement of connectivity estimates from ultra-high angular resolution diffusion MRI, neuroanatomical insight provided by polarised light imaging and myelin-stained histology, and the joint analysis of MRI and microscopy data for reconstruction of the microscopy-inspired connectome. All data and code are made openly available.

Our ability to characterise brain connectivity has been greatly advanced by the scientific community's open access to big data. Big data can employ large cohorts to examine both inter- and intra-subject variability (in e.g. the UK Biobank[1] or Human Connectome Project[2]), or aim to characterise a single brain in exquisite detail[3]. Here we introduce "The BigMac Dataset" —**Big** data in a **Mac**aque brain—a resource that combines in vivo MRI with extensive postmortem MRI and whole-brain, multi-contrast microscopy data in a single macaque brain. Big-Mac consists of in vivo data acquired over multiple sessions, over 270 h

postmortem MRI scanning, over 1000 h of microscopy data acquisition and several terabytes of raw data.

The BigMac dataset combines multimodal data from both MRI and microscopy to explicitly address issues of sensitivity and specificity in MRI. MRI is a powerful non-invasive method that can inform on whole-brain structure and function, which can in turn be related to cognition, behaviour or medical outcomes. However, MRI also faces several limitations. In vivo signals are typically noisy and confounded by artefacts due to physiological effects and technical bottlenecks related to

[1]Wellcome Centre for Integrative Neuroimaging, FMRIB Centre, Nuffield Department of Clinical Neurosciences, University of Oxford, Oxford, UK. [2]Division of Clinical Neurology, Nuffield Department of Clinical Neurosciences, University of Oxford, Oxford, UK. [3]Wellcome Centre for Integrative Neuroimaging, Experimental Psychology, Medical Sciences Division, University of Oxford, Oxford, UK. [4]Department of Oncology, University of Oxford, Oxford, UK. [5]Donders Institute for Brain, Cognition and Behaviour, Radboud University Nijmegen, Nijmegen, The Netherlands. [6]These authors contributed equally: Saad Jbabdi, Karla L. Miller. ✉e-mail: amy.howard@ndcn.ox.ac.uk

hardware limitations or the requirement for short scan times. Furthermore, MRI signals are often an indirect measure of the brain features of interest, making interpretation challenging. Diffusion MRI maps the microscopic motion of water molecules as they randomly move through tissue, to infer structural connectivity or changes in cellular morphology. These analyses require complex computational signal modelling with many strong assumptions about how the tissue microstructure relates to the diffusion signal, which if inaccurate, can bias model outputs. Crucially, the measurements are averaged over millimetres of tissue and so inference on brain structure or function at the micrometer scale is hard, if not ill-posed. Consequently, characterisation of the connectome using MRI alone faces significant limitations.

Alternatively, connectome data can be acquired via light microscopy, which is frequently used to study brain structure with micrometre or sub-micrometre resolution. Typically, thin tissue sections are processed to visualise specific cellular structures where this high 'specificity' approach has applications from basic neuroanatomy to disease mechanisms. Microscopy is however often limited to interrogating small, ex vivo tissue sections and thus has limited applications in vivo. Nonetheless, when microscopy is combined with MRI, it affords the opportunity for multi-scale neuroscience, interconnecting microscopic cellular processes with macroscopic MRI signals.

Such research ideally requires data that (i) combines complementary MRI and microscopy data; (ii) relates high-quality postmortem MRI and microscopy to in vivo MRI; (iii) facilitates whole brain analysis with densely sampled data throughout; (iv) has co-registered MRI and microscopy data to facilitate meaningful voxelwise comparisons, and (v) provides the above in a single specimen. The latter is essential as only when combining data from the same brain can we be sure that we are not over generalising, or ignoring important inter-subject variations. Few existing open datasets fulfil all these requirements. For example, existing data combining MRI and microscopy in the same tissue sample are often limited to small tissue sections[4–6], a single microscopy contrast or minimal MRI data[7], or compare data from different tissue samples, overlooking considerable between-brains variability. The BigMac dataset aims to address each of these goals, combining co-registered in vivo MRI, extensive postmortem MRI and whole-brain multi-contrast microscopy data in a single macaque brain.

The BigMac dataset is interesting from both an anatomical and methodological viewpoint. For those interested in microstructural neuroanatomy, the densely sampled multi-contrast microscopy can be used to examine both the myelo- and cyto-architecture in great detail and develop novel atlases or parcellations. For those interested in multi-scale neuroscience, BigMac provides whole brain multimodal data spanning four orders of magnitude with which we can link microscale cellular structures to macroscale features of brain organisation and function. For those interested in diffusion modelling, the comprehensively sampled diffusion MRI space—which is co-registered to microscopy data—can be used to drive protocol optimisation, advance computational modelling of the tissue microstructure or brain connectivity, and provide direct validation of many current and future diffusion MRI models and analysis methods[8,9]. Finally, for those interested in machine- or deep-learning approaches, the BigMac dataset will support the development of novel algorithms which jointly model MRI and microscopy data[10,11], for example, to map quantitative imaging biomarkers to specific features of the tissue microstructure.

This paper documents the first release of the BigMac dataset. The open data includes in vivo and postmortem MRI, as well as whole brain microscopy data from polarised light imaging[12–14] and myelin-stained histology[15], both of which provide detailed information about tissue myeloarchitecture[16]. Here we detail the multi-faceted data acquisition and curation, and conduct some of the first analyses which demonstrate the data quality and unique information or analyses afforded by BigMac.

## Results

First we provide an overview of the data included in BigMac. We then explore the BigMac data from various viewpoints including (i) how ultra-high angular resolution diffusion imaging affects connectivity estimates, (ii) how microscopy can be used to detail the myeloarchitecture of the brain, (iii) the quality of MRI-microscopy co-registration, (iv) voxelwise comparisons of MRI and microscopy metrics, and (v) a method for performing hybrid MRI-microscopy tractography. Here we primarily analyse the postmortem data in BigMac, where translating our work to the in vivo domain is the focus of future work.

### Data summary

Figure 1 gives an overview of the BigMac dataset, with further acquisition details provided in Table 1. Importantly, the BigMac dataset includes in vivo data combining behavioural data with diffusion MRI (1 mm isotropic resolution), task fMRI (2 mm isotropic), resting-state fMRI (2 mm isotropic) and structural MRI (0.5 mm isotropic) acquired over multiple scan sessions[17,18]. This provides an excellent opportunity to link high-quality postmortem MRI and microscopy data (detailed structural connectivity) to in vivo data (structural and functional connectivity) within the same brain.

Ex vivo, a comprehensive MRI dataset (~270 h scanning time) was acquired which includes high resolution (0.3 mm isotropic) structural images and extensive diffusion MRI data at two spatial resolutions (0.6 and 1 mm isotropic). The 0.6 mm data includes 128 gradient directions at $b = 4$ ms/μm², whilst the 1 mm data includes 250 gradient directions at $b = 4$ ms/μm² and 1000 gradient directions at $b = 7$ and 10 ms/μm² ('ultra-HARDI data'). This ultra-HARDI data is complemented by diffusion data with spherical tensor encoding at b-values of $b = 4, 7$, and 10 ms/μm² for advanced microstructural imaging. Images with negligible diffusion weighting ($b \sim 0$ ms/μm²) and a $T_1$ map were also acquired. The postmortem MR data in BigMac serves two specific aims. First, the postmortem data acts as a crucial intermediary between in vivo diffusion MRI and postmortem microscopy: the postmortem and in vivo MRI are similar in nature (i.e. both are MRI signals) but image the tissue in different states (postmortem, fixed tissue versus in vivo), whereas the postmortem MRI and microscopy share a similar tissue state, though the signals are different. Second, compared to in vivo MRI, the postmortem data is of particularly high quality. It benefits from being acquired on a small-bore preclinical scanner at higher field strength with exceptionally long scan times, and avoids signal instabilities from physiological movement that can fundamentally limit the signal-to-noise ratio (SNR).

After scanning, the entire brain was sectioned along the anterior-posterior axis, with consecutive sections allocated to one of six microscopy contrasts (Table 1). These sections were sequentially allocated to polarised light imaging, to visualise the orientation of myelinated fibres[12–14], histological staining of myelin (Gallyas silver) or Nissl bodies (Cresyl violet), or kept for complementary staining that is to be decided. These unassigned sections were returned to formalin and stored for longevity. The PLI and histology-stained slides were then imaged at high resolution (4 or 0.28 μm/pixel) and co-registered to the MRI[19].

Through this extensive, multimodal, multi-contrast acquisition, the BigMac dataset aims to provide a detailed characterisation of the macaque connectome, where the data and analysis tools are openly disseminated to the scientific community (c.f. Data and Code Availability).

### Ultra-high angular resolution diffusion imaging

The postmortem diffusion MRI protocol had three main objectives (Table 5.1): (i) high angular resolution imaging (a), (ii) high spatial resolution imaging (b), and (iii) combining linear and spherical tensor encoding (c). The diffusion MRI data includes two spatial resolutions: high resolution 0.6 mm data at $b = 4$ ms/μm² as well as 1 mm data at

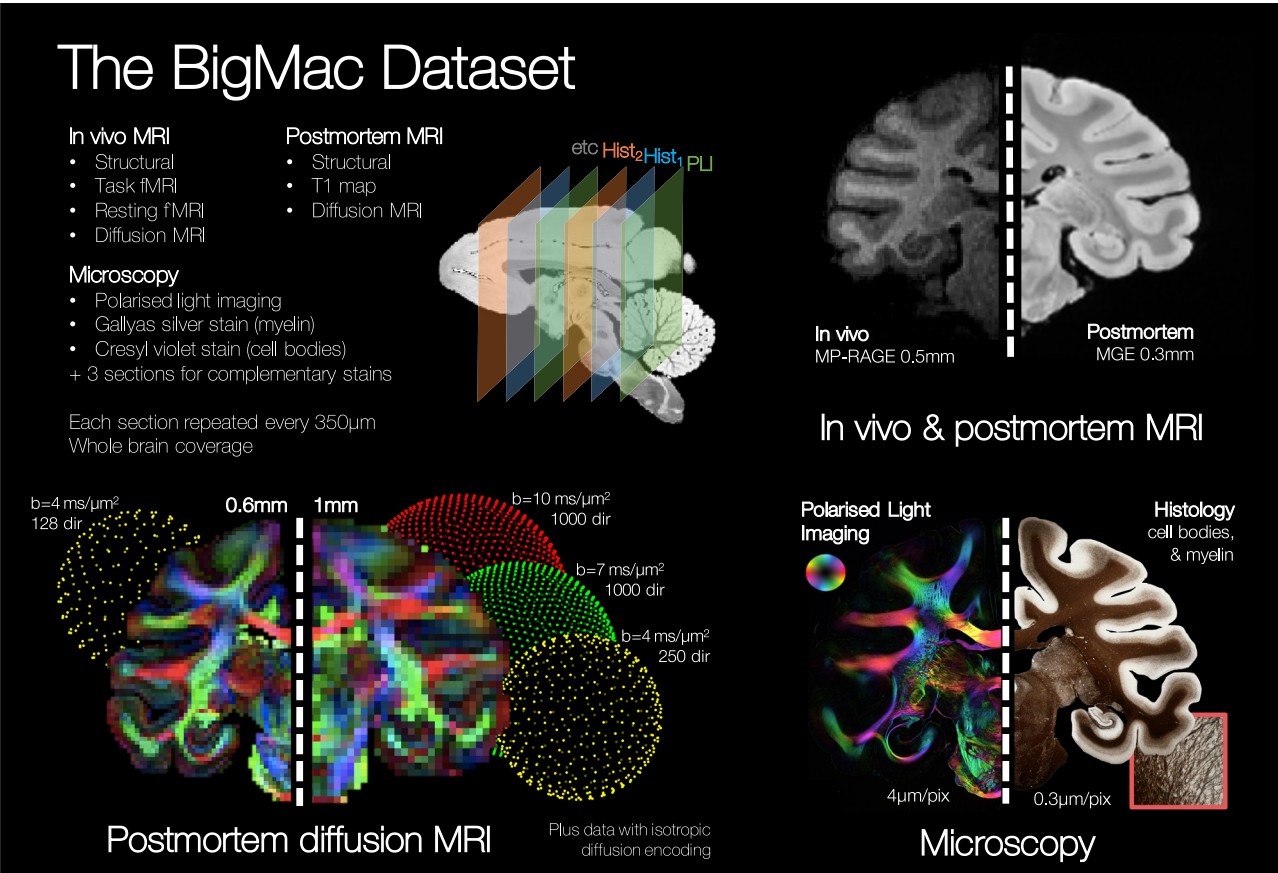

**Fig. 1 | A summary of the data in the BigMac dataset.** A single macaque monkey was scanned both in vivo and postmortem. The in vivo MRI data includes structural, task functional MRI (fMRI), resting fMRI and diffusion MRI. The postmortem diffusion MRI data contains up to 1000 gradient directions for b-values of 4, 7 and 10 ms/μm² and two spatial resolutions (0.6 and 1 mm isotropic). A $T_1$ map and data with spherical tensor encoding were also collected. For comparison, polarised light imaging and histology provide high-resolution information about the tissue microstructure at 4 μm/pix and 0.28 μm/pix respectively. In total, the postmortem MRI acquisition constituted around 270 h scanning time and the microscopy data took >1000 h to acquire.

b-values of 4, 7 and 10 ms/μm² (Fig. 1). Here the 0.6 mm protocol was designed to strike a good balance between spatial and angular resolution. In comparison, the 1 mm protocol facilitates the characterisation of complex fibre geometries through a more extreme sampling of q-space, with multiple *b*-values—where the reduced resolution provides increased SNR at high *b*—and either 250 or 1000 gradient directions per shell. In addition, at 1 mm we were able to acquire 1000 gradient directions ('ultra-HARDI data') in the outer two shells within an achievable scan time (<1 week). Figure 2a shows the diffusion signal from a single voxel, demonstrating how with 1000 gradient directions we are able to examine the 3D signal profile in great detail.

We first examined whether having high angular resolution provided unique microstructural information with regards to fibre orientations and structural connectivity. Here, the Ball and Stick model[20,21] was used to estimate the fibre orientations from data acquired at different b-values (*b* = 4, 7 or 10 ms/μm²) and varied angular resolution (64 – 1000 gradient directions). Examining white matter voxels, Fig. 2b shows how as the angular resolution of the data was increased, the Ball and Stick model estimated an increasing number of voxels with multiple fibre populations (Fig. 2b top) and that the fibre orientations were estimated with increased precision (Fig. 2b bottom). Both effects were most striking in the tertiary fibres, suggesting that high angular resolution data can resolve less dominant fibre populations which are overlooked in lower angular resolution data, and which could have important functional or structural profiles.

Differences in the estimated fibre populations from the Ball and Stick model from high angular resolution data may have downstream effects on tractography-based structural connectivity estimates. Here we hypothesised that increased angular resolution should result in superior fibre tracking through crossing fibre regions, improving the reconstruction of long range connections. Using probabilistic tractography (probtrackX[21]) we reconstructed a dense connectivity matrix for the 82 cortical regions of interest (ROIs) included in the Kötter and Wanke Regional Map (RM) parcellation[22–24]. Figure 2c shows the fractional increase in the number of streamlines from the matrix with 1000 gradient directions and that with 64 gradient directions ($(N_{1000} - N_{64})/N_{64}$). The ultra-high angular resolution dataset estimates more streamlines reaching almost every pair of ROIs. There is a particularly high number of additional streamlines between many interhemispheric connections (green box). A notable exception to this overall pattern is that parietal and premotor regions (coded yellow-orange on the depicted brain surface) do not show a large increase in interhemispheric connectivity with higher angular resolution (turquoise box) but do show increased connections to other regions in the same hemisphere (blue box). Many homotopic regions (where we expect there to be true connections) have a > 1.8-fold increase, where we tend to see a larger effect in more lateral homotopic regions. Those with a > 50-fold increase include the ventral part of the anterior visual area (VACv), the inferior temporal cortex (TCi) and the central temporal cortex (TCc), (see Supplementary Figs. 1–3 for a more detailed discussion of these results). The high angular resolution data produced notably longer streamlines, with a > 5-fold increase in the number of streamlines whose length was >50 mm.

**Table 1 | A summary of the postmortem MRI and multi-contrast microscopy data in the BigMac dataset**

| Postmortem MRI | Resolution (mm iso) | b-value (ms/µm²) | # directions (linear) | # averages (spherical) | # b=0 volumes | δ/Δ (ms) | TE/TR (ms/s) |
|---|---|---|---|---|---|---|---|
| Structural | 0.3 | - | - | - | - | - | 7.8 / 0.097 |
| T$_1$ maps | 0.6 | - | - | - | - | - | 8 / 10 |
| Diffusion MRI | 0.6 | 4 | 128 | - | 8 | 7/13 | 25.4 / 10 |
| Diffusion MRI | 1 | 4 | 250 | - | 10 | 14/24 | 42.5 / 3.5 |
| Diffusion MRI | 1 | 7 | 1000 | - | 40 | 14/24 | 42.5 / 3.5 |
| Diffusion MRI | 1 | 10 | 1000 | - | 40 | 14/24 | 42.5 / 3.5 |
| Diffusion MRI | 1 | 4 | - | 30 | 1 | - | 42.5 / 6.4 |
| Diffusion MRI | 1 | 4 | 50 | - | 2 | 14/24 | 42.5 / 6.4 |
| Diffusion MRI | 1 | 7 | - | 30 | 1 | - | 42.5 / 6.4 |
| Diffusion MRI | 1 | 7 | 50 | - | 2 | 14/24 | 42.5 / 6.4 |
| Diffusion MRI | 1 | 10 | - | 30 | 1 | - | 42.5 / 6.4 |
| Diffusion MRI | 1 | 10 | 50 | - | 2 | 14/24 | 42.5 / 6.4 |

a) High spatial resolution, b) Ultra-high angular resolution (ultra-HARDI), c) Combining linear & spherical tensor encoding

| Microscopy | Thickness (µm) | Staining | Visualisation | Imaging resolution |
|---|---|---|---|---|
| Polarised Light Imaging | 50 | None | Myelinated fibres | 4 µm/pix |
| Histology | 50 | Cresyl violet | Nissl bodies | - |
| Histology | 50 | - | - | - |
| Histology | 50 | Gallyas silver | Myelin | 0.28 µm/pix |
| Histology | 100 | - | - | - |
| Unstained | 50 | - | - | - |

The postmortem diffusion data includes three protocols which can broadly be described as achieving a) high spatial resolution, b) ultra-high angular resolution, and c) combining linear and spherical tensor encoded data. The first data release includes PLI and myelin-stained histology (Gallyas); additional histological stains will be added to the BigMac resource as these data are acquired. *iso* isotropic; # number; *DW* diffusion-weighted; *TE* echo time; *TR* repetition time; 'linear' and 'spherical' indicate diffusion MRI acquired with linear and spherical tensor encoding.

Finally, Fig. 2d compares tractography reconstruction of the superior longitudinal fasciculus (SLF) II from data with high angular resolution (top), low angular resolution but high b-value (middle) or high spatial resolution (bottom). Although this tract is not difficult to reconstruct in human data, it's reconstruction can be challenging in the macaque. Here anatomically constrained probabilistic tractography was performed using XTRACT[25] with predefined seed, target and exclusion masks. In Fig. 2d we use a fairly high threshold (0.01 = 1%) on the normalised tract density mask to isolate the tract centre (i.e. the voxels with the highest density of streamlines). In the ultra-HARDI data, the high angular resolution and contrast allows us to reconstruct the tract with a single, uniform core. We observe a higher density of streamlines extending the main tract body into the frontal and posterior regions, following the expected spatial extent and connectivity of the SLF II. Neither the tract from lower angular or high spatial resolution data have the same anterior-posterior reach. Instead these tracts appears to have a systematic false positive offshoot (yellow arrows) extending to the superior cortex, which may be indicative of streamlines crossing to the SLF I[26].

Together, these results suggest that higher angular resolution data may be advantageous when trying to reconstruct long range or inter-hemispheric connections which likely track through crossing fibre regions.

**Microscopy characterisation of brain myeloarchitecture**
After scanning, the whole brain was sectioned along the anterior-posterior axis, with consecutive sections allocated to either polarised light imaging[12–14] or one of five histology contrasts. Each microscopy contrast includes ~200 slides spanning the entire brain, with 350 µm separation between consecutive slides with the same contrast. As microscopy imaging takes many hundreds of hours, the first data release includes PLI and myelin-stained histology (Gallyas[15]) as detailed below; additional histological stains will be added to the BigMac resource as these data are acquired.

Polarised light imaging (PLI)[12–14] utilises the birefringence of myelin to estimate the primary fibre orientation per microscopy pixel. Figure 3a shows example PLI from different locations along the anterior-posterior axis where image 1 is the most anterior and image 8 the most posterior. Here we see the myelinated fibres in high detail at a resolution of 4 µm per pixel. The orientations are colour-coded in HSV (hue-saturation-value) space, where the hue is dependent on the fibre orientation and the value is related to the tissue birefringence. The PLI data can, for example, track white matter fibres fanning across the cortex (3a, i), bundles projecting between deep grey matter structures such as the globus pallidus and the putamen (ii), as well as pontocerebellar fibres extending from the basilar sulcus and around the pons (iii). Further the PLI can differentiate cortical layers which differ in the orientation and density of myelinated axons ((i), white arrow).

In the cerebellum (3a,iv) we see i) clear delineation of the dentate nucleus (white arrow) and ii) how the PLI data separates the molecular and granular layers of the cerebellar cortex[27]; the molecular layer is the top, outermost layer of the cerebellar cortex (Fig. 3b green arrows) and the granular layer (yellow arrows) lies between the molecular layer and the white matter. Notably, in many regions the molecular layer shows fairly strong birefringence, with in-plane angles parallel to the cortical surface. In comparison, the molecular layer at the gyri crown (blue arrows) tend to have low birefringence. Although the birefringence in the granular layer is low, we can observe individually identifiable fibres

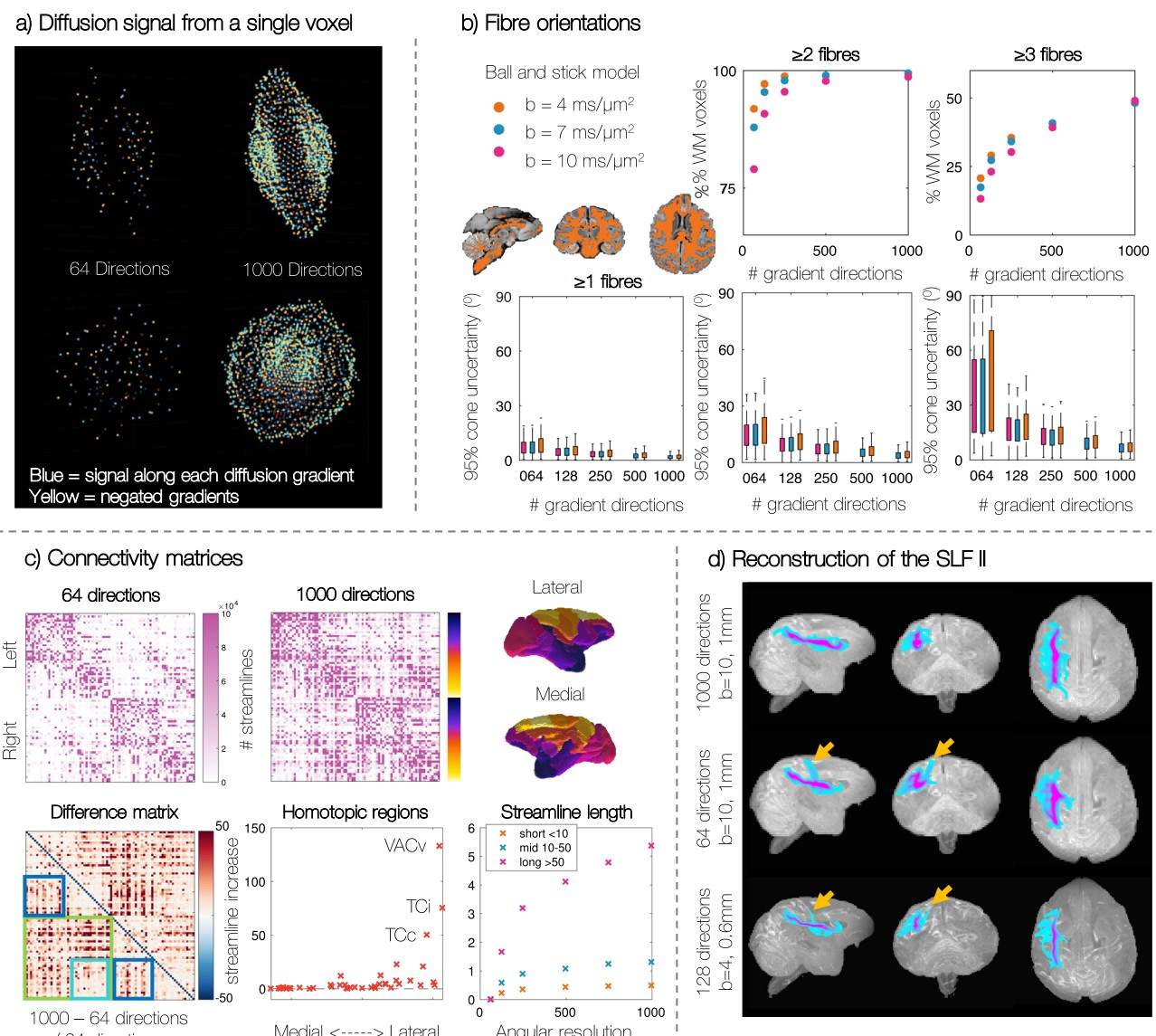

**Fig. 2 | Ultra-high angular resolution postmortem diffusion MRI allows us to (a) characterise the diffusion signal in great detail, (b) find more tertiary populations and estimate fibre orientations with increased precision, (c) reconstruct longer streamlines and increase inter-hemispheric connectivity, particularly to the occipital and temporal lobe, and (d) reconstruct the SLF II with increased confidence. a** Each point represents the diffusion signal along a single gradient direction. The voxel is shown from two orthogonal views. **b** The Ball and Stick model was fitted to data with varying angular resolution. Both the number of fibre populations, and the precision of the orientation estimates is plotted. Boxplot interpretation: box limits indicate upper and lower quartiles, whilst whiskers indicate the range of data not considered as outliers. The white matter mask included 30061 voxels for the $b = 4$ ms/μm$^2$ and 30690 voxels for $b = 7$ or 10 ms/μm$^2$ data. **c** Comparing connectivity matrices from 64 and 1000 gradient datasets. Since the difference matrix is symmetric, the bottom half is used to highlight interesting features: the green box indicates interhemispheric connectivity, with the turquoise box showing connections between parietal and premotor areas; the dark blue boxes show intrahemispheric connectivity of areas that are separated along the inferior-superior axis. Streamlines whose length <10 mm are considered `short', 10–50 mm are `mid' length and >50 mm are `long'. **d** Reconstruction of the superior longitudinal fasciculus (SLF) II using probabilistic tractography.

or fibre bundles which generally extend from the white matter and into the molecular layer. This pattern is observed in many but not all areas of the cerebellar cortex.

To hypothesise on the origin of the birefringence in the molecular layer, we should consider the known architecture of the cerebellar cortex (Fig. 3b right)[27]. The granular layer (between the white matter and the molecular layer) contains cerebellar granule cells, some of the smallest but most numerous cells in the brain. The axon of the granule cell extends vertically into the molecular layer, where it then splits into two horizontal branches in a 'T-like' fashion. These branches are known as 'parallel fibres'. The Purkinje cells sit with their soma in the 'Purkinje layer', at the interface of the granular and molecular layer, and the

Purkinje dendritic tree extends into the molecular layer. The parallel fibres of the granule cells run through the Purkinje dendrites, forming synaptic connections with the Purkinje dendritic spine. Consequently, it seems reasonable that these parallel fibres, running in line with the cortical surface, could cause the coherently oriented birefringence of the molecular layer. Similar conclusions were reached by Koike-Tani et al.[28] who attributed high birefringence in the molecular layer of a late stage chick embryonic cerebellum to the presence of densely packed, non-myelinated parallel fibres.

Although PLI signal from brain tissue is typically associated with the myelin, histological data did not support the presence of myelinated fibres in the molecular layer (Fig. 3b left). Furthermore,

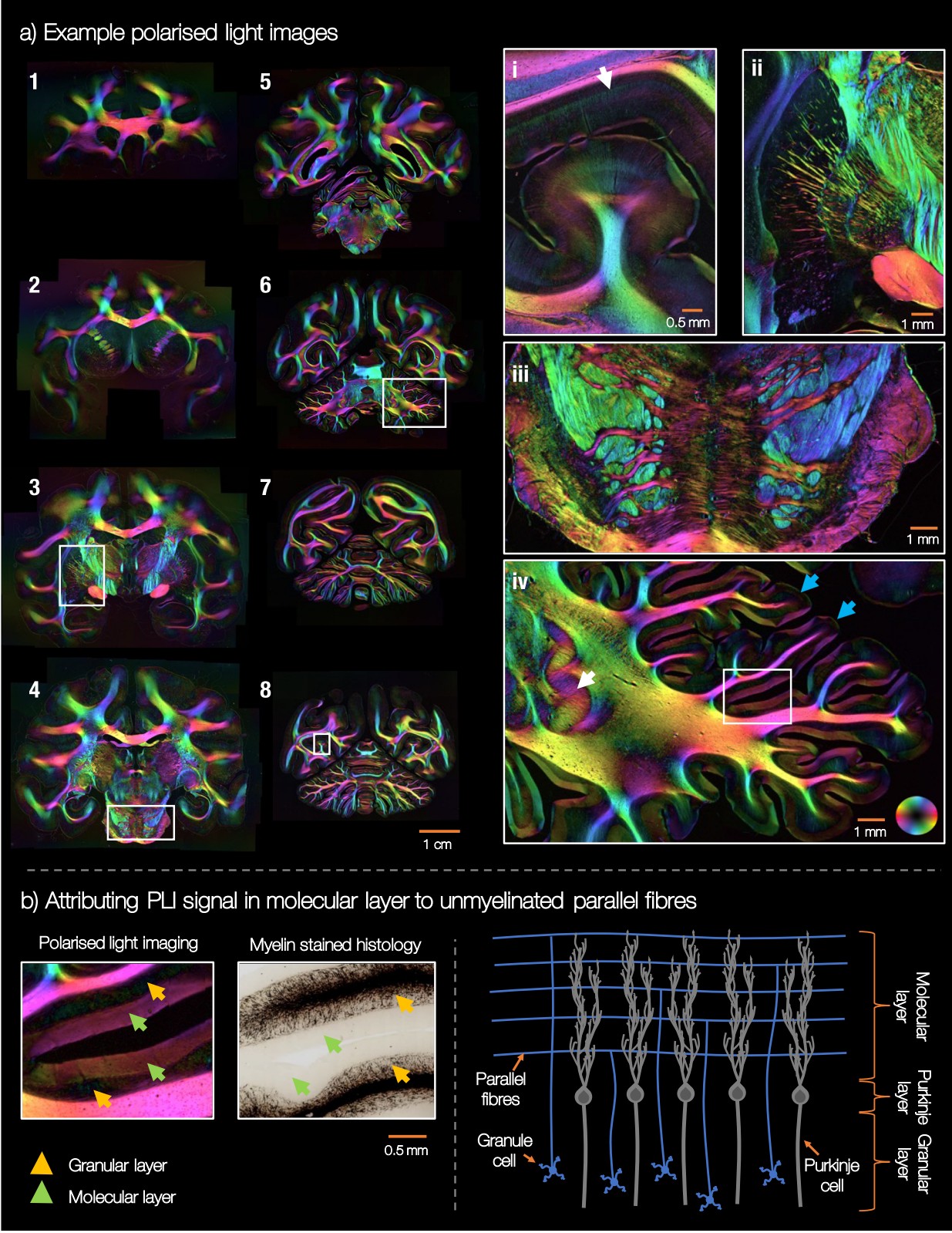

a) Example polarised light images

b) Attributing PLI signal in molecular layer to unmyelinated parallel fibres

Polarised light imaging

Myelin stained histology

▲ Granular layer
▲ Molecular layer

Parallel fibres

Granule cell

Purkinje cell

Molecular layer

Purkinje layer

Granular layer

non-myelinated fibres often exhibit positive birefringence with respect to the longitudinal axis of the fibre due to the presence of axon organelles and myelin membrane proteins, rather than negative birefringence associated with the lipid bilayer of more heavily myelinated fibres[29,30]. Such positive birefringence would result in PLI orientations orthogonal to those observed[30], as the "fast axis" of the medium output from our PLI analysis would now indicate the radial rather than

longitudinal axis of the fibre. Here there could be two effects: (i) that the PLI signal is sensitive to not only myelin, but also other aspects of microstructure orientational coherence[31], or (ii) that the parallel fibres have some small amount of myelin which is detected by PLI but not picked up by the histology stain. Indeed, though parallel fibres are often assumed to be exclusively unmyelinated[32], using electron microscopy Wyatt et al.[33] found myelinated fibres of 0.4–1.1 μm

**Fig. 3 | Polarised light imaging in BigMac.** The colours correspond to orientations described by the 2D colour wheel in image (iv). Note, this is different from the standard 3D colour representation in diffusion MRI. **a** Example PLI throughout the brain. Image 1 = most anterior, 8 = most posterior of the images shown. The myeloarchitecture is viewed in great detail due to the 4 μm / pixel PLI resolution. We observe fibres projecting into the cortex (right hand panel part (i)), into and through subcortical structures (ii), around the pons (iii) and across the cerebellum (iv). Inset (i) comes from image 8, (ii) from image 3, (iii) from image 4 and (iv) from image 6. In D, the blue arrows point to the gyri crown. **b** We hypothesise that the PLI signal in the cerebellar molecular layer can be attributed to parallel fibres. **b** (left) The yellow and green arrows point to the granular layer and the molecular layer respectively. **b** (right) The structure of the cerebellar cortex. This highly simplified schematic focuses solely on the granule and Purkinje cells, to illustrate the parallel fibres in the molecular layer. Note, the dendritic tree of the Purkinje cells has highly anisotropic dispersion. Here we see the axis of least dispersion, where the Purkinje dendritic tree fans out most in the through-page orientation. PLI were acquired for 192 slides throughout the brain.

diameter in the macaque molecular layer. Interestingly, Wyatt et al. found a larger number of myelinated fibres in the molecular layer near the Purkinje layer, and fewer towards the cortical surface. Consequently, we might expect to see a gradient in the PLI signal across the molecular layer, though this is not evident in the data. Future work is required to fully understand the origin of this birefringence in the molecular layer, though Fig. 3 provides some evidence for PLI sensitivity to coherently orientated, anisotropic structures irrespective of their degree of myelination[28,34], that may be omitted from classic myelin histology. Simulations of the PLI signal, for example based on the *fastPLI* framework[35], may help provide insight.

Figure 4 a shows example images from myelin-stained histology (Gallyas silver,[15]) included in the anterior BigMac brain. We see interesting detail in both the white and grey matter where we can visualise single, thin axons in detail due to both the sensitivity of the stain and the sub-micrometre imaging resolution (0.28 μm/pixel). In the digitised images we can track complex patterns of fibre projection for example, from the white matter into the cortex (i), between subcortical structures (ii), within the highly complex geometry of the hippocampus (iii) or through the deep white matter (iv). Furthermore, in the deep white matter we see different tissue 'textures' as well as large scale, 'wave-like' undulations of fibres in the corpus callosum (iv) (see Supplementary Fig. 4 for an enlarged image). Though this a region which is often considered coherently ordered with little fibre dispersion, these data corroborate previous observations of fibre dispersion or incoherence in the corpus callosum[4].

The myelin-stained slides were analysed using structure tensor analysis[36–39] to estimate the primary fibre orientation per microscopy pixel (Fig. 4b). Figure 4c shows the primary fibre orientation derived from the structure tensor analysis per ~40 μm superpixel. The orientations are colour-coded in HSV (hue-saturation-value) space similar to PLI. Here, pixels in heavily stained white matter are bright compared to those in the lightly stained grey matter as the "value" was set to $(1 − r)$, where $r$ is the grey-scale pixel intensity. This image is then compared to an adjacent section imaged with PLI (Fig. 4d). Despite the very different manner by which the orientation estimates were derived, the two methods provide corroborating information in both the white matter, where the myelin stain is very dense precluding the identification of individual fibres, and the cortex, where myelinated fibres are less dense. Furthermore, we see how the two modalities provide subtly different information. For example, when a PLI pixel (at 4 μm/pix resolution) contains crossing fibre populations of approximately equal weighting and perpendicular orientations, the PLI-derived in-plane angle becomes uninformative, the PLI signal is low and the HSV image dark. In Fig. 4e (right) we see how this results in a darker bands appearing between fibre tracts of different orientations. For example, where the pink and teal fibres intermingle, as indicated by the white arrows. In these crossing fibre regions, the histology data has an order of magnitude finer spatial resolution (at 0.28 μm/pixel) and so can likely resolve both fibre populations, the mean of which is shown in Fig. 4b, c, e.

**Co-registration of MRI and microscopy data**
Our ability to meaningfully compare MRI and microscopy data is greatly enhanced by having high quality multimodal data registration, like that provided in BigMac. However, co-registration of the BigMac MRI and microscopy data was highly challenging for a number of reasons. Firstly, the spatial resolution of the data spans up to 4 orders of scale, from microscopy images with sub-micrometre (histology) or micrometre (PLI) resolution to the ~ millimetre MRI data. Secondly, the contrast between the MRI-microscopy images is substantially different and may highlight different tissue features. For example, in both the Gallyas and PLI images, neither the grey/white matter boundary nor outer tissue edges are consistently well defined. Thirdly, the thin tissue sections may be deformed during sectioning or microscopy pre-processing (e.g. staining and/or mounting). For example, the tissue can shrink or tear, or there may be dirt or bubbles in the microscopy slides. Finally, the inherently 2D microscopy must be registered into the 3D volume - a particularly difficult optimisation with many degrees of freedom.

Figure 5 shows an example BigMac registration using TIRL, a new MRI-microscopy registration tool by Huszar et al.[19] that is specifically designed to overcome the above challenges and facilitate accurate MRI-microscopy co-registration. In BigMac, TIRL generated the mapping between microscopy and the structural MRI, and FSL tools (FLIRT/FNIRT[40,41]) were used for cross-modality registration within MRI (e.g. co-registering structural and diffusion MRI). TIRL utilises a sequence of linear and non-linear transforms to register 2D microscopy images into 3D MRI volumes. Further, the TIRL cost function is based on a modality independent neighbourhood descriptor (MIND)[42], which is explicitly designed to capture correspondence in spatial information in a way that is agnostic to image contrast. By concatenating the TIRL and FSL transforms, the TIRL platform could then be used to map the high-resolution microscopy pixels into the 3D MR volume in any MR domain, or MR voxels onto a 2D microscopy plane. Figure 5b shows the quality of the TIRL transform where the MRI data maps very closely to the microscopy image, both at the white/grey matter boundaries (outlined in green) and the tissue edge (in orange). Additional TIRL outputs are provided in Supplementary Fig. 5.

**Comparing fibre orientations from microscopy and MRI**
With co-registered MRI and microscopy data, BigMac facilitates voxelwise comparisons of quantitative microstructural metrics extracted from both modalities. Figure 6a shows example co-registered maps from postmortem MRI and microscopy, where the DTI and myelin-histology have been registered to the PLI reference image. The primary axis of the diffusion tensor ($V_1$)[43] has been projected onto the PLI plane and uses the same 2D colour map as the PLI. We observe excellent agreement between $V_1$ and the orientations derived from polarised light imaging. This facilitates direct validation of orientation information from MRI against microscopy, as illustrated in Fig. 6b. Here, the diffusion signal has been modelled using a biophysical model, the Ball and Rackets model (BAR)[44], to extract a single, disperse fibre orientation distribution per voxel, which has then been projected onto the microscopy plane. This is compared to co-registered fibre orientation distributions from both PLI and Gallyas-stained histology. We observe good correspondence of the dominant fibre orientations in both coherent regions such as the corpus callosum and more complex relationships in crossing fibre regions such as the centrum semiovale. Here both orientational information, and the amount of fibre

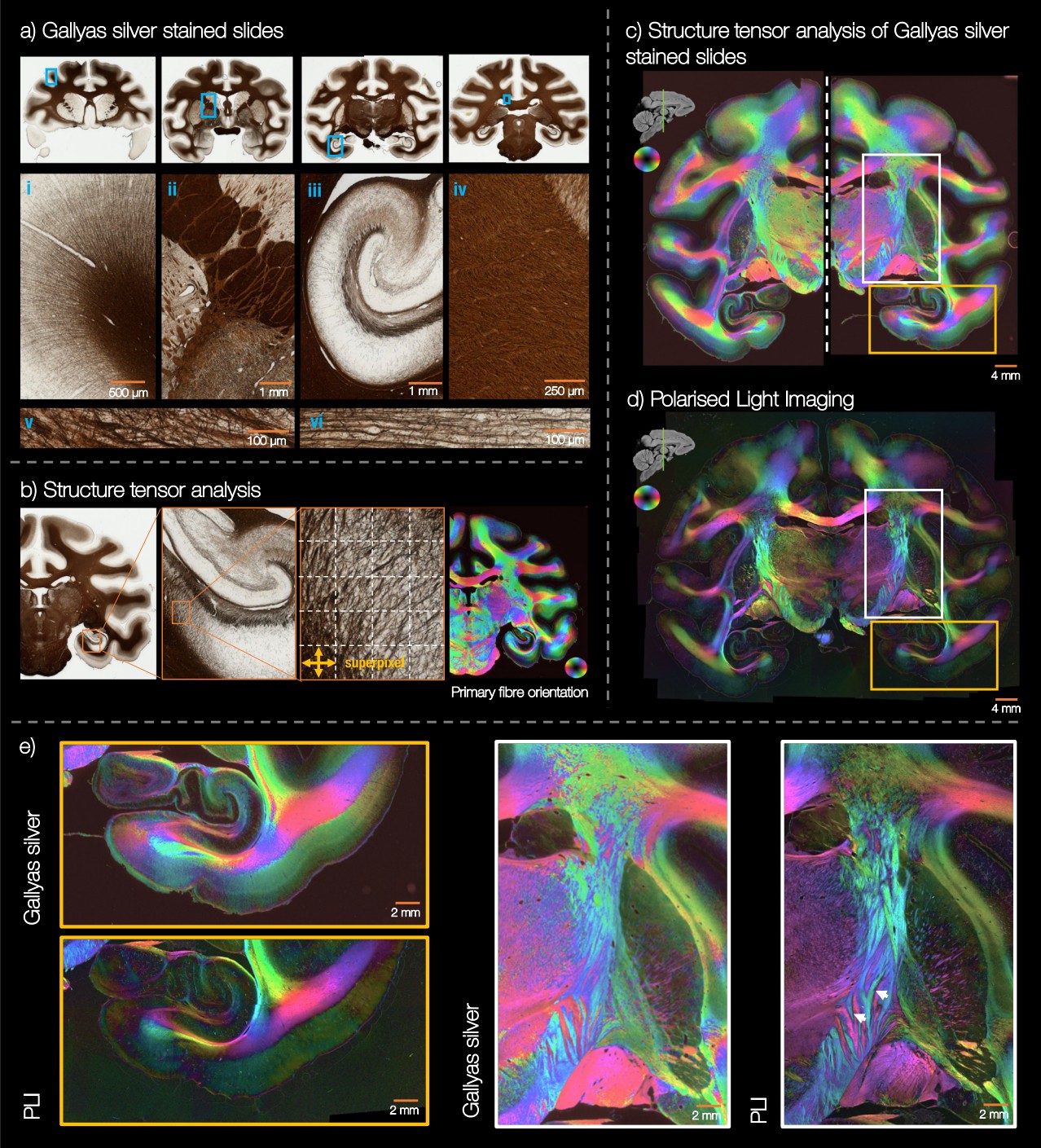

**Fig. 4 | Gallyas silver stained histology in BigMac. a** Example digitised slides. With a spatial resolution of 0.28 μm/pixel, we see the myelinated fibres is great detail (i–vi), visualising single axons at the grey/white matter boundary (v–vi) and fibre undulations in the deep white matter (iv). **b** Structure tensor analysis was applied to the Gallyas silver stained slides to estimate a fibre orientation per 0.28 μm pixel. **c**, **d** The fibre orientations derived from the Gallyas silver stained slide are compared to an adjacent PLI slide. We see remarkable consistency between the images with both modalities capturing the myeloarchitecture in detail. **e** Both modalities show the detailed organisation of the hippocampus and surrounding white/grey matter (yellow box), as well as the corticospinal tract (white box). We acquired 197 myelin stained slides in total: 120 slides from the anterior brain with the same quality staining as presented in a,b, and 77 slides from the posterior brain which contain some artefact as outlined in Supplementary Fig. 8. All slides were processed using structure tensor analysis.

dispersion can be compared. Indeed, the slightly higher levels of dispersion away from the midline of the corpus callosum may be related to the 'wave like' fibre patterns observed in the myelin-stained histology (Fig. 4d and Supplementary Fig. 4).

The fibre dispersion can then be quantified using the orientation dispersion index (ODI), which ranges from 0 for perfectly aligned fibres, to 1 for isotropic dispersion[45]. Figure 6c compares dispersion estimates across many white matter voxels (top, covering 20 consecutive PLI and histology slides each), and a subset of voxels from the centrum semiovale (bottom). We see fair correspondence between the dispersion from myelin-stained histology and the diffusion model. Estimates of dispersion from PLI appear less reliable, in line with

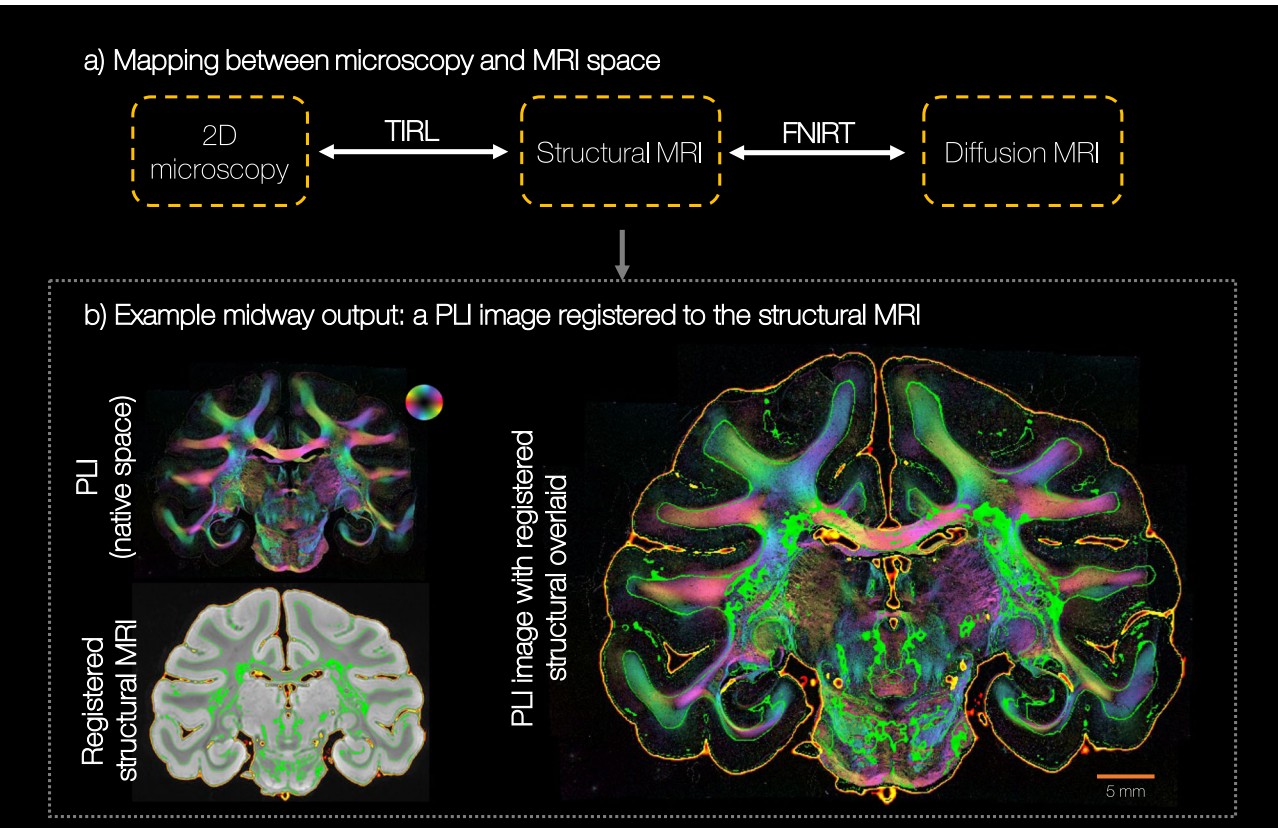

**Fig. 5 | Co-registration of the BigMac MRI and microscopy data. a** Each microscopy slide was independently registered to the postmortem structural MRI (MGE), using multi-scale MRI-microscopy registration (TIRL). Other MR modalities (here diffusion MRI) were co-registered to the postmortem structural MRI using MRI-MRI linear and non-linear registration (FLIRT/FNIRT) respectively. By combining the two warpfields, the microscopy data can be directly transformed into any MRI space or vica versa. **b** An example output of TIRL for a single polarised light image. The structural image is clipped to indicate the tissue outline (orange) and white/grey matter boundary (green). These outlines are then overlaid on the PLI image (right). Co-registration was performed for all 192 PLI slides and 197 myelin-stained slides.

previous reports[4] and observations in 6b. This may be related to PLI estimating a single orientation per PLI pixel which likely includes many axons (axon diameter is typically ~ 1 μm[46], whilst each PLI pixel with an in-plane resolution of ~ 4 μm and a slice thickness of 50 μm covers a volume of ~ 800 μm³). In comparison, the histology data has an order of magnitude higher in-plane resolution (0.28 μm, covering a volume of ~4 μm³) which may lead to a more faithful estimate of fibre dispersion.

As well as comparing MRI-microscopy equivalents, microscopy can be used to understand indirect relationships with MR parameters. For example, histology dispersion is shown to have a clear negative correlation with fractional anisotropy (FA) from DTI (Fig. 6d, ref. 43). In comparison, dispersion has a weak but significant ($p = 9 \times 10^{-14}$) correlation with microscopic FA ($\mu$FA)[47,48] in the white matter (top) and no significant correlation in the centrum semioval (bottom), a known deep white matter region of complex dispersion. This is reassuring as the $\mu$FA parameter is explicitly meant to be independent of the fibre orientation distribution: in the centrum semiovale, $\mu$FA is independent of dispersion, where the small negative correlation across all of white matter may be driven by partial volume effects. In future work when additional microscopy contrasts are added to BigMac, multivariate regressions can be performed to better understand how complex tissue microstructure relates to sensitive but not specific diffusion metrics such as those from the diffusion tensor or other signal models[49].

The regression models in Fig. 6c, d reached significance (estimated $p < 0.001$ where $p < 0.05$ is considered significant) for all plots except for 6d bottom left comparing histology ODI with $\mu$FA ($p = 0.23$,

hence regression model is not shown). Similar relations were found when correlating in vivo diffusion MRI estimates of fibre dispersion and FA with ODI from both PLI and histology (Supplementary Fig. 6).

**Towards the microscopy connectome: hybrid MRI-PLI tractography**

One of the primary limitations of the microscopy data in BigMac is that it only informs on the fibre orientations in the 2D plane of sampled slides, precluding 3D reconstruction of the microscopy connectome. In comparison, the diffusion data can provide orientational information in 3D, but with limited spatial resolution (0.6–1 mm isotropic). Figure 7 demonstrates one approach to joint modelling[10] where we combine in-plane orientations from microscopy (here PLI) with through-plane information from postmortem diffusion MRI (dMRI) to reconstruct 3D 'hybrid dMRI-PLI fibre orientations' at the resolution of the microscopy data. For each PLI pixel, we compare the PLI in-plane orientation to orientations estimated from co-registered diffusion data using the Ball and Stick model[20] which have been projected onto the PLI plane (Fig. 7a). The PLI through-plane angle is then approximated by that from the most similar Ball and Stick orientation. This produces a hybrid dMRI-PLI orientation that is both 3D and at the resolution of the microscopy data. These orientations can then be combined into a hybrid fibre orientation distribution (FOD) which can be directly compared with those from diffusion MRI and input into existing tractography algorithms for tract reconstruction. Here we reconstruct hybrid orientations using the PLI data in BigMac, though our current method is also applicable to histology slides analysed using structure tensor analysis.

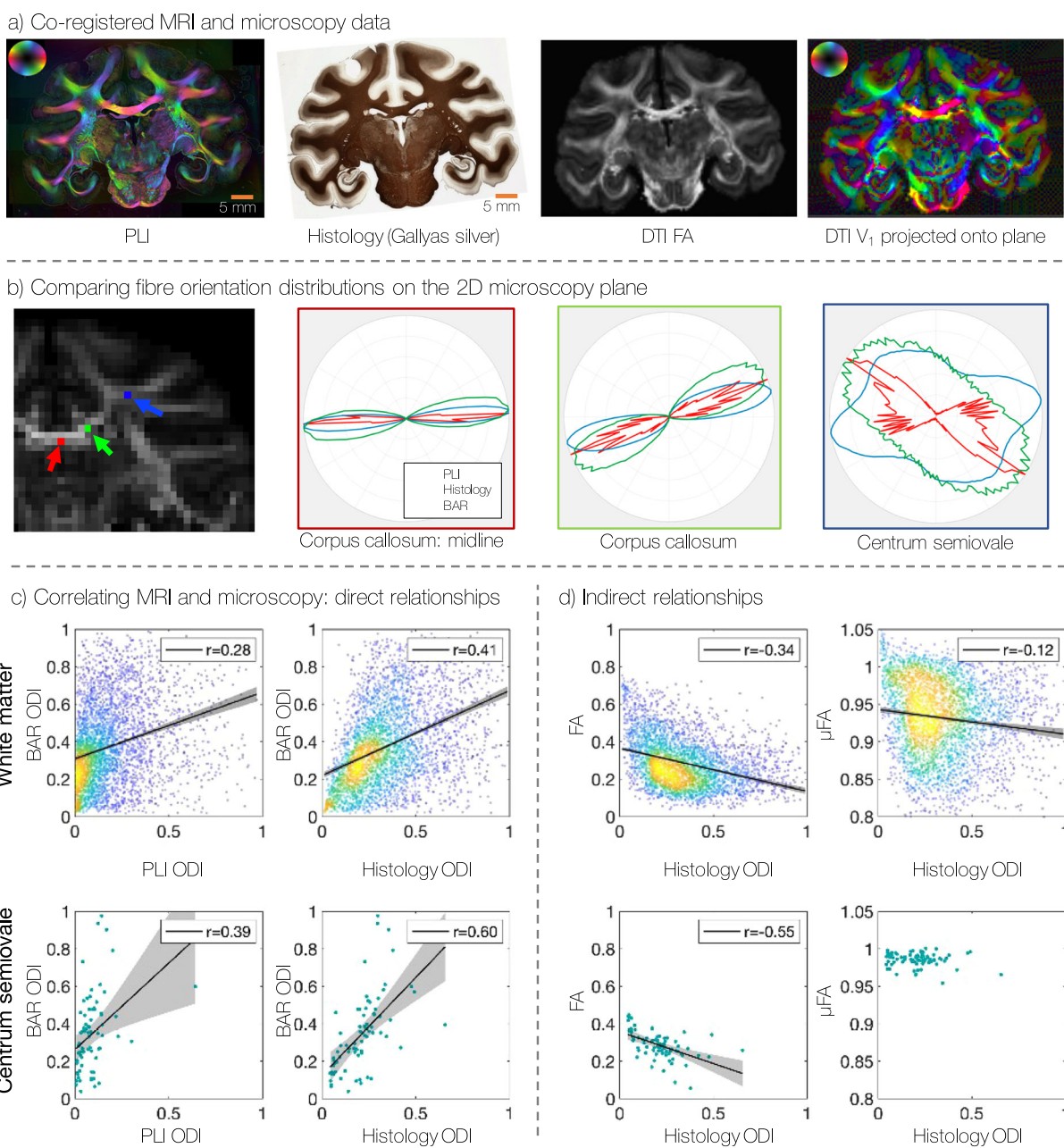

**Fig. 6 | Comparing information from co-registered postmortem diffusion MRI and microscopy. a** Co-registration facilitates qualitative comparisons of microscopy and MRI (here DTI data) when both are warped to a common space. We see clear correspondence between the PLI orientations and the primary eigenvector of the diffusion tensor that has ben projected onto the microscopy plane. **b** A 2D fibre orientation distribution is extracted on a voxelwise basis from both PLI, myelin-stained histology, and the Ball and Rackets model (BAR) for diffusion MRI ($b = 10\ ms/\mu m^2$, 1 mm). **c** The orientation dispersion index (ODI) from microscopy is correlated with various diffusion metrics: the ODI from the Ball and Rackets model, fractional anisotropy (FA) from the diffusion tensor model (both calculated from $b = 10\ ms/\mu m^2$, 1 mm data), and $\mu$FA from data with multiple tensor encodings ($b = 4, 7$ and $10\ ms/\mu m^2$, 1 mm). The top row shows data points from a white matter mask with ~ 3700 voxels (mask shown in Supplementary Fig. 6) where blue-yellow indicates a low-high density of points in the scatter plot. The bottom row shows only a subset of voxels in the centrum semiovale (84 voxels). In **c**, **d**, the black lines show the line of best fit, the grey lines indicate the 95% confidence intervals, and $r$ is the correlation coefficient.

Figure 7b shows example dMRI-PLI FODs at varying spatial resolutions. Reassuringly, the hybrid FODs show smoothly varying patterns in all three dimensions, even when the hybrid FODs are of higher spatial resolution than the diffusion MRI (0.6 mm hybrid reconstruction versus 1 mm diffusion data). Interestingly we observe notably fewer voxels with crossing fibre populations than we might expect from diffusion MRI in regions such as the centrum semiovale. The hybrid FODs can then be reconstructed at very high in-plane resolutions ($\gtrsim 4 \times 4\ \mu m$ in-plane, $\gtrsim 350\ \mu m$ through-plane) to observe fine structural details, or investigate the effect of spatial resolution on FOD reconstruction or downstream tractography (the subject of future work). As proof of concept, the corticospinal tract was reconstructed from the hybrid FODs using anatomically constrained tractography (Fig. 7c). Future work will consider whole brain reconstruction of the 3D microscopy connectome at high spatial resolution. We expect this microscopy-inspired connectome to both provide new anatomical insight, and be a valuable resource for validating and advancing in vivo tractography[9].

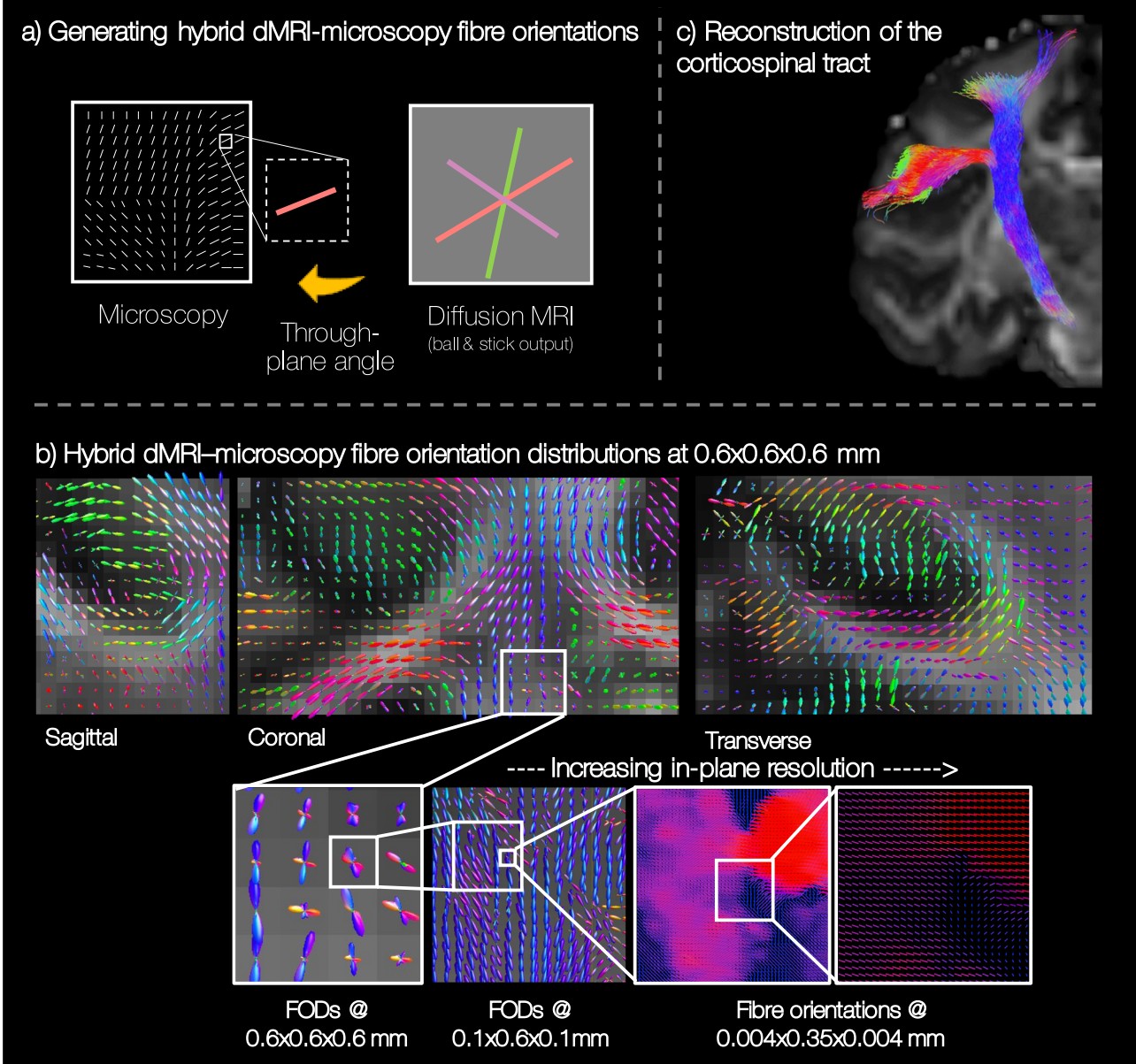

**Fig. 7 | Estimating hybrid diffusion MRI-PLI fibre orientations for future reconstruction of the microscopy-inspired connectome (see text for details). a** Where microscopy (here PLI) informs on fibre orientations within the microscopy plane, the through plane was approximated by that from postmortem diffusion MRI (Ball and Stick model[20], $b = 10$ ms/µm², 1 mm). **b** The hybrid orientations were combined into fibre orientation distributions at increasing in-plane resolutions. **c** The hybrid FODs can be input into pre-existing tractography methods to successfully reconstruct in white matter bundles such as the corticospinal tract.

## Discussion

The BigMac dataset aims to characterise a single connectome in exquisite detail, combining MR signals with high resolution microscopy data throughout the macaque brain. The postmortem diffusion MRI includes high b-value ultra-HARDI data to characterise the diffusion signal in great detail, estimate white matter fibre orientations with high precision, and improve the reconstruction of white matter tracts through crossing fibre regions. Furthermore, with 1000 gradient directions we retain dense sampling on an arbitrary 2D plane for direct comparison with 2D microscopy data. The microscopy includes coronal slides of polarised light imaging[12–14] and myelin-stained histology that has been densely sampled throughout the brain. This high-resolution, high specificity data allows us to visualise tissue myeloarchitecture in detail to provide neuroanatomical insight - such as the orientational coherence of the PLI signal in the cerebellar molecular layer - and act as a pseudo ground truth estimate of tissue

microstructure against which MR metrics can be compared. Crucially, the MRI and microscopy data have been carefully co-registered, facilitating novel data fusion analyses including the hybrid diffusion MRI-microscopy tractography presented here.

The dataset has several limitations including: the 2D nature of the microscopy, which precludes 3D visualisation of the cell bodies or nerve fibres within each microscopy section; the multimodal nature of the microscopy prohibits whole-brain 3D reconstruction of the tissue microstructure from a single contrast at high resolution (as in e.g. the BigBrain dataset[7]); tissue processing artefacts (*c.f. Methods: Polarised light imaging*) and inconsistent staining (*c.f. Methods: Gallyas silver staining*) in some PLI and histology slides respectively; and the choice of MR data acquired where here we chose to comprehensively sample q-space in the long diffusion time regime, rather than, for example, acquire data at multiple diffusion times for sensitivity to restricted compartments, or acquire MRI at very high spatial resolution. The

latter could be overcome by acquiring additional MRI on a different monkey brain that was subsequently co-registered to the BigMac data, though such analyses would have to assume some level of anatomical consistency between brains. Finally, we acknowledge that this data was acquired from a single macaque brain, with lesions in the orbitofrontal cortex and an abnormality in the left hemisphere. This may limit the datasets suitability for specific applications.

Nonetheless, as a unique, multimodal resource that complements existing open data (for cross-modality, cross-subject or cross-species investigations, as well as data from invasive macaque studies not possible in humans), the BigMac dataset will enable neuroscientists to ask new and fundamental questions. For example, it will enable researchers to (i) link brain connectivity across spatial scales spanning four orders of magnitude; (ii) understand the relationship between relatively crude but non-invasive MRI signals and cellular properties measured by invasive microscopy; and (iii) develop novel approaches to extract rich information from in vivo MRI data by leveraging the specificity of microscopy-derived gold standards. The BigMac dataset provides an open access platform from which we can interconnect microstructural features with MRI signals throughout the brain. Details on how to access the data, further documentation and code are provided in the Data and Code Availability sections below.

Future work will include characterising the unique information provided by the hybrid MRI-PLI tractography, as well as completing ongoing microscopy work to add Nissl-stained histology and other complementary stains (e.g neurofilament protein SMI32 or glia Iba-1 or GFAP) to the BigMac resource.

## Methods

### Data acquisition

At the centre of the BigMac dataset is the brain of a single adult rhesus macaque (*Macaca mulatta*, male). The animal was cared for, and data were acquired by, researchers at the University of Oxford, UK. All procedures were performed under licences from the United Kingdom (UK) Home Office in accordance with the UK Animals (Scientific Procedures) Act 1986 and with European Union guidelines (EU Directive 2010/63/EU).

During it's adult life, the macaque was scanned in vivo over multiple scan sessions. At 11.7 years of age, the brain was perfusion fixed in formalin after which extensive postmortem MRI data was acquired. After scanning, the entire brain was sectioned along the anterior-posterior axis, with consecutive slices processed for different microscopy contrasts. The acquisition timeline for presented data was as follows:

- In vivo MRI session 1: April 2010 (4 years old)
- In vivo MRI session 2: January 2017 (10 years 9 months old)
- Perfusion fixation: December 2017 (11 years 7 months old)
- Postmortem MRI: March-April 2018 (3–4 months postmortem)
- Postmortem microscopy: 2018 onwards

**Tissue pathology.** As part of a behavioural study (in preparation), the BigMac monkey underwent intraoperative bilateral lesioning of the orbitofrontal cortex. The postmortem MRI data in Supplementary Fig. 7 shows the extent of the bilateral lesions -1 year after surgery.

In addition to the planned lesion, inspection of the postmortem data (Supplementary Fig. 8) shows a fairly substantial abnormality in the left hemisphere which extends from the inferior portion of the supramarginal gyrus up through the post-central sulcus. This abnormality could relate to a cerebral bleed, which perhaps occurred post operatively, though no behavioural or other observations were made that would relate to this abnormality.

### In vivo MRI

Prior to sacrifice, the animal partook in a number of studies[17,18] in which behavioural and imaging data were acquired. One study[17] combines

functional MRI with a decision-making task to investigate the of role surprising events (i.e. prediction errors) on reward-based learning. A second[18] links flexible behaviour to changes in both the MRI-derived structure and function of a fronto-cortical network.

The BigMac in vivo MRI data includes structural images, diffusion MRI, resting-state fMRI and task fMRI over a variety of tasks. The data were acquired at various time points throughout the animal's adult life. The in vivo data were acquired on a 3 T whole-body scanner (Gmax = 40 G/cm) with a four-channel phased-array receive coil and a local transmit coil (Windmiller Kolster Scientific). Here we include data acquired at two separate time points. "Session 1" includes diffusion, structural and resting-state fMRI, where complementary resting-state and structural data from another 19 animals has previously been made openly available through the PRIMatE Data Exchange (PRIME-DE) for cross-subject comparisons (c.f. Data Availablity). "Session 2" includes similar MRI from a shorter acquisition that occurred only 1 year before sacrifice (the last in vivo scan). As such, the age-induced atrophy between the in vivo and postmortem data should be roughly similar. During scanning the animal was kept under minimum anaesthetic using similar procedures to those previously described[50–52].

Structural MRI images were acquired using a $T_1$-weighted Magnetization Prepared—RApid Gradient Echo (MP-RAGE) sequence with 0.5 mm isotropic resolution, TE/TR = 4.01 ms/2.5 s and 128 slices. Whole brain fMRI data (BOLD) were acquired with echo planar imaging (EPI) and 2 mm isotropic resolution: TE/TR = 19 ms/2 s, 1600 volumes for Session 1 and 800 volumes for Session 2. This corresponds to 52 min 26 s and 26 min 13 s of data respectively. Diffusion MRI data were acquired using EPI with 1 mm isotropic resolution, TE/TR = 100 ms/8.2 s and a $b$-value of 1 ms/μm². 1100 diffusion weighted (81 unique gradient directions) and 144 non-diffusion weighted volumes were acquired with both ± phase encoding directions for Session 1, and 361 diffusion weighted (61 unique gradient directions) and 38 non-diffusion weighted volumes were acquired with both ± phase encoding directions for Session 2. The data were distortion corrected using pipelines from the MR comparative anatomy toolbox (MrCat) and FSL tools[53,54]. Additional task fMRI maps (z-stats) are available via[17,18].

To indicate the quality of the in vivo and postmortem data, Supplementary Fig. 9 (top) shows example structural images from the most recent in vivo imaging session. This is then compared to postmortem data, as described below.

### Postmortem MRI

At 11.7 years of age the animal was anaesthetised and the brain perfusion fixed with 90% saline and 10% formalin, extracted and then stored in 30% sucrose formalin. Postmortem data were then acquired on an 7 T small animal scanner (Agilent) fitted with a 40 G/cm gradient coil (Agilent, 205/120 mm) and a Birdcage receive/transmit RF coil (Rapid Biomedical, 72 mm). Prior to scanning the brain was rehydrated in phosphate-buffered saline to remove lasting fixative and somewhat restore both the diffusivity and $T_2$ of the tissue[55,56]. The brain was then packed into a plastic holder filled with Fluorinert (FC-3283, 3 M™, St. Paul, USA), a proton-free, susceptibility-matched fluid which is MR invisible and improves field homogeneity.

As the diffusion properties of brain tissue are highly dependent on the tissue temperature[57], the temperature was controlled by passing air at the constant temperature of 20°C.

The BigMac postmortem MRI data was acquired over three different scanning sessions:

1. 29th March–5th April 2018: Acquisition of $b$ = 7 and 10 ms/μm² ultra-HARDI data with 1000 diffusion-weighted gradient directions per shell and 1 mm isotropic resolution.
2. 6th–9th April 2018: Acquisition of the 0.3 mm structural MRI, $T_1$ mapping and 0.6 mm $b$ = 4 ms/μm² diffusion-weighted data.
3. 20th - 23rd April 2018: Acquisition of 1 mm $b$ = 4 ms/μm² diffusion-weighted data plus the protocol combining linear and spherical

tensor encoding at $b = 4, 7$ and $10 \, \text{ms/µm}^2$. The brain was not repacked between sessions, though slight deformations did occur as the tissue relaxed over time. In total, the postmortem MRI data acquisition took ~ 270 h scanning time. All postmortem data were corrected in 3D for Gibbs ringing artefacts (mrdegibbs3D, MRtrix[58–60]) prior to other preprocessing.

**Structural MRI.** Two structural images were acquired with subtly different contrast: one a with multi gradient echo (MGE 3D) sequence, and one using balanced steady-state free procession (bSSFP).

The MGE parameters were: TE/TR = 7.8/97.7 ms, flip angle = 30°, 0.3 mm isotropic resolution, FOV = 76.8 × 76.8 × 76.8 mm. The structural image was subsequently corrected for bias field and segmented using FAST[53,54,61]. The white and grey matter masks were then hand edited to provide precise segmentation of the white and grey matter.

The bSSFP data were acquired using a TRUFI sequence with 16 frequency increments: TE/TR = 3.05/6.1 ms, flip angle = 30°, 0.3 mm isotropic resolution, FOV = 76.8 × 76.8 × 76.8 mm. The structural image was formed by averaging the data using root-mean sum of squares.

The Supplementary Fig. 9 (bottom) shows example postmortem structural images from the BigMac dataset. Note how the contrast is inverted when related to the in vivo T1-weighted images. Here we purposefully acquire T2/T2*-weighted postmortem data as conventional T1w typically don't give good contrast postmortem due changes in relaxation times. Due to their high image quality and anatomical detail, the postmortem structural MRI act as a crucial intermediary in the co-registration of both the diffusion MRI and microscopy data (c.f. Co-registration) and the in vivo and postmortem MRI.

**$T_1$ mapping.** A $T_1$ map was acquired using similar imaging parameters as the 0.6 mm postmortem diffusion MRI data but now with an inversion recovery preparation: TE/TR = 8.6 ms/10 s, FOV = 76.8 × 76.8 × 76.8 mm, resolution 0.6 mm isotropic and 12 inversion times (TI) from 10 to 6000 ms. The Barral model $S(\text{TI}) = a + b \exp{(-\text{TI}/T_1)}$[62], where $S$ is the MR signal and $[a, b, T_1]$ are unknowns, was fitted voxelwise to the data to obtain quantitative estimates of $T_1$ (inversion_recovery, qMRLab[63]).

**Diffusion-weighted MRI.** The diffusion-weighted data were acquired using a spin echo multi-slice (DW-SEMS) sequence and single-line readout. To ensure that data from different shells retain the same diffusion propagator, both the time between the gradients (i.e. the diffusion time, $\Delta$) and gradient duration ($\delta$) were kept constant for all data with 1 mm isotropic resolution. The desired b-value was achieved by modifying the amplitude of magnetic gradient, $G$.

### High spatial or angular resolution

In the 1 mm diffusion data with high angular resolution, data were acquired in batches of 26 volumes where one volume with negligible diffusion weighting ($b \sim 0 \, \text{ms/µm}^2$) was followed by 25 diffusion-weighted volumes. Two sets of gradient directions were used: one with 250 gradient directions ($b = 4 \, \text{ms/µm}^2$), the other with 1000 gradient directions ($b = 7, 10 \, \text{ms/µm}^2$). For both sets, the gradient directions were generated using GPS (an FSL tool,[64]) and were evenly distributed across the sphere. The directions were then ordered so that any consecutive subset of gradient directions (e.g the first 100 gradient directions) also gave good coverage across the sphere (orderpoints, Camino[65]). In this case, were the scan interrupted or prematurely stopped, we would retain reasonable angular coverage. Finally, to evenly spread the heating of the magnetic gradients, the gradient directions within each batch of 25 were reordered to ensure that highly co-linear directions were not played out in close succession.

The 1 mm data acquisition parameters were as follows: TE/TR = 42.4 ms/3.5 s; FOV = 76 × 76 × 76 mm; $\delta/\Delta = 14/24$ ms; 1 mm isotropic resolution; time per gradient direction = 4.4 min; $b = 4 \, \text{ms/µm}^2$ data had $G = 12.0 \, \text{G/cm}$, 250 gradient directions and 10 non-diffusion weighted volumes; $b = 7 \, \text{ms/µm}^2$ had $G = 15.9 \, \text{G/cm}$, 1000 gradient directions and 40 non-diffusion weighted volumes; $b = 10 \, \text{ms/µm}^2$ had $G = 19.1 \, \text{G/cm}$, 1000 gradient directions and 40 non-diffusion weighted volumes.

The 0.6 mm $b = 4 \, \text{ms/µm}^2$ data followed a different protocol. Here 128 diffusion-weighted gradient directions were acquired, followed by 8 volumes with negligible diffusion weighting. The acquisition parameters were as follows: TE/TR = 25.4 ms/10 s; FOV = 76.8 × 76.8 × 76.8 mm; $\delta/\Delta = 7/13$ ms; time per gradient direction = 21.3 min; $b = 4 \, \text{ms/µm}^2$; 0.6 mm isotropic resolution; $G = 32 \, \text{G/cm}$.

### Preprocessing

The postmortem MRI data was found to have few distortions, so minimal preprocessing was applied. For example, the data did not need correcting for susceptibility or eddy current distortions. This is largely due to the brain being placed in a susceptibility-matched fluid and the data acquired with a single-line readout instead of the typical echo planar imaging (EPI). The main corrections were (a) registration (both within and between session), (b) correction of signal drift, and (c) signal normalisation.

### Registration

The ultra-HARDI data for both $b = 7$, and $10 \, \text{ms/µm}^2$ were acquired within the first scanning session. At specific time points throughout the week-long acquisition, the central scanner frequency was recalibrated. This occurred three times during the $b = 7 \, \text{ms/µm}^2$ acquisition and 4 times during $b = 10 \, \text{ms/µm}^2$. Because of the recalibration, images acquired with different scanner central frequencies are shifted (translated) with respect to one another. To correct for these translations, the data were rigidly registered to a reference $S_0$ image (i.e. a volume with negligible diffusion weighting) from the ultra-HARDI dataset. Here the reference image was taken to be the mean $S_0$ image from the first 'set' of images which were all acquired with the same central frequency. The registration was performed using FLIRT with spline interpolation of the data[40,66].

Data from the second scan session includes high spatial resolution (0.6 mm isotropic) $b = 4 \, \text{ms/µm}^2$ diffusion data as well as the detailed (0.3 mm isotropic) structural scan. Upon inspection, the $S_0$ images associated with the 0.6 mm diffusion data ($b = 4 \, \text{ms/µm}^2$) appeared to slowly drift in position along the readout direction. To correct for signal drift, the $S_0$ images were linearly registered and intensity normalised to the first $S_0$ i.e. that which most likely represents the 'true' $S_0$ of the diffusion-weighted data. The data were aligned using FLIRT[40,66] where the transformation was restricted to only consider translation along a single axis. The $b = 4 \, \text{ms/µm}^2$ 0.6 mm diffusion-weighted data were then co-registered to the postmortem structural image using linear registration (FLIRT,[40,66]).

Supplementary Figure 10 describes how data acquired in different sessions was registered together using either linear or non-linear transforms (FLIRT/FNIRT[40,41,64,66]). Data acquired within the same scan session were registered using linear transforms. Upon inspection of the data, the brain shape appeared to change or 'relax' slightly between scanning sessions. To account for these deformations, non-linear transformations were generated both between the $b = 4 \, \text{ms/µm}^2$ 1 mm data and the ultra-HARDI data, and between the ultra-HARDI data and the postmortem structural image[41,64]. Consequently, data users should take care to account for voxelwise rotations in the gradient directions according to the non-linear warpfield when combining non-linearly registered diffusion data from different shells.

Finally, to integrate the BigMac dataset with other datasets, non-linear transformations[41,64] were computed between the postmortem structural image and the F99 standard template[67]. Here we utilise a $T_1$-like image, created from the structural MRI using hand-edited white and grey matter masks, because non-linear registration requires

images with similar contrast and the BigMac ex vivo structural image has inverted contrast when compared to the in vivo F99 $T_1$.

*Signal drift*

In all experiments, the signal magnitude, measured as the mean signal across $S_0$ images, was seen to decrease over time. To correct for signal drift, a linear trend with respect to time was fitted to the $S_0$ images and subsequently regressed from the data (both the $S_0$ and diffusion-weighted volumes).

*Data normalisation*

Most diffusion models approximate the $S_0$ image by taking the mean $S_0$ image across all volumes with minimal diffusion weighting, assuming that the signal magnitude is constant across time. In contrast, here we found the $S_0$ signal to vary between scanning sessions and b-shells. Were diffusion models naively applied to concatenated data from the BigMac dataset, the results may be biased (e.g. the kurtosis would be misestimated). Consequently, the diffusion-weighted data was normalised to the mean $S_0$ of the $b = 10$ ms/µm² ultra-HARDI data.

**Combining linear and spherical tensor encoding**

Combining data with linear and spherical tensor encoding allows for the separation of effects due to the fibre orientation distribution and the diffusion properties, to estimate additional microstructural parameters such as $\mu$FA[47,48]. In BigMac, data with spherical tensor encoding were also acquired at b-values of 4, 7 and 10 ms/µm². The gradient waveform was optimised using the NOW toolbox in Matlab[68]. Due to additional stress on the magnetic gradients when performing the spherical tensor encoding, the repetition time (TR) was increased with respect to the ultra-HARDI data: TE/TR = 42.5 ms/6.4 s; FOV = 76 × 76 × 76 mm; 1 mm isotropic resolution. For each b-value, 30 images were acquired with spherical tensor encoding and 1 with negligible diffusion weighting. Complementary data with linear tensor encoding and the same TR were also acquired: 50 gradient directions per shell with $\delta/\Delta = 14/24$ ms, plus 2 volumes with negligible diffusion weighting. The gradient amplitude $G$ was adjusted to produced the required b-values of $b = 4$, 7 and 10 ms/µm².

Data were corrected for Gibbs ringing and signal drift as above. All data were normalised to the linear tensor encoded $b = 10$ ms/µm² mean $S_0$. Maps of $\mu$FA as well as isotropic and anisotropic kurtosis were generated following the DIVIDE framework and fitting the Laplace transform of the gamma distribution (*dtd_gamma* model) using the multi-dimensional MRI toolbox (md-dmri)[47,69,70].

**Microscopy**

Using the BigMac dataset, we can link the MRI signal to microscopy data which has both micrometre resolution and high specificity. In BigMac, the brain was sectioned, stained, and imaged ('processed') in two batches. The brain was first cut around the level of the posterior tip of the central sulcus to create two tissue blocks, representing the anterior half and posterior half. First, the anterior block was sectioned, stained, and imaged, after which the posterior block was processed using a highly similar protocol.

Each tissue block was sectioned on a frozen microtome along the anterior-posterior axis to produce thin coronal tissue sections. Consecutive sections were allocated, in order, to one of six contrasts:

1. Polarised light imaging to visualise myelinated fibres (50 µm thick)
2. Cresyl violet staining of Nissl bodies (50 µm thick)
3. Unassigned section (50 µm thick)
4. Gallyas silver staining of myelin (50 µm thick)
5. Unassigned section (100 µm thick)
6. Unassigned section (50 µm thick)

Each contrast was repeated every 350 µm throughout the brain. The unassigned sections were returned to formalin and stored for longevity.

The imaging of the tissue sections is very time consuming. Hence, slide digitisation is an ongoing process where the Nissl and other complementary stains will be released at a future date.

**Polarised light imaging.** Polarised light imaging (Fig. 3) utilises the birefringence of myelinated axons to estimate the primary fibre orientation per pixel[12-14]. Here unstained tissue sections were imaged using a Leica DM4000B microscope with an automated stage (Leica, Germany, using Leica Application Suite X software) adapted for PLI with an LED light source, a polariser, a quarter wave plate with its fast axis at 45 degrees to the transmission axis of the polariser, and a rotatable polariser (the analyser).

Due to the large size of the BigMac tissue sections, multiple fields of view were acquired across each sample and later stitched together to form a whole slide 'mosaic' image. For each field of view, images were taken as the analyser was rotated from 0 to 180 degrees in 9 equidistant steps. A 2.5x magnifying objective produced an imaging resolution of ~4 µm per pixel. PLI processing was performed using in-house developed MATLAB scripts[4]. Background correction was performed[4,71] to account for light source variations across the image, after which a sinusoid was fitted to the pixelwise image intensity as a function of the analyser rotation. Maps of transmittance, retardance, and in-plane angle were derived from the sinusoid phase and amplitude[12-14].

Supplementary Figure 11 shows example PLI mosaics from the BigMac dataset. The transmittance map is related to the amount of light extinguished by the sample. The retardance map is dependent on both the inclination and amount of birefringent material (i.e. myelinated fibres) within the PLI pixel[12-14]. In the HSV image, the hue is dependent on the in-plane angle of the myelinated fibres, and the value is given by the tissue retardance.

By assuming that the amount of myelin is approximately constant across the white matter, an inclination angle can be estimated from the retardance map. This inclination estimate relies on knowing the myelin thickness and birefringence, which here was set to a somewhat arbitrary, constant value. Consequently, the estimated inclination angles are likely inaccurate and should not be used as a quantitative microscopy metric. In this work, the "inclination" map is used solely for MRI-PLI co-registration.

The anterior PLI sections were mounted using a hard-set mounting medium (FluorSave, Merck) where over time we saw artefacts (bubbles) develop on the slides. This artefact is observed in the PLI transmittance image (Fig. 5), though the retardance and in-plane maps do not appear to be substantially affected in the white matter apart from faintly visible edge effects (white arrows). In some anterior PLI sections we see background birefringence outside of the tissue which varies slowly across the slide (Fig. 3 sections 1–5 where 1 and 3 are worst affected). This is due to the slides being coated in a small amount of gelatine which aids the mounting of tissue sections onto glass slides but which is also birefringent[72]. Nonetheless, the PLI orientations within the white matter do not appear greatly affected, where the birefringence of the myelin appears to dominate[30]. The posterior sections (which were processed second) were instead mounted with an aqueous mounting medium (Polyvinylpyrrolidone, PVP) on plain glass slides without gelatine coating.

**Gallyas silver staining.** Gallyas silver staining[15] was used for histological visualisation of the myeloarchitecture[16] (Fig. 4). In this method, colloidal silver particles bind to myelin and turn deep brown. After staining, the sections were cover-slipped, sealed and digitised using a Aperio ScanScope Turbo AT slidescanner (Leica) with a 20x/0.75 NA Plan Apo objective lens coupled with an x2 optical magnification lens to achieve a total magnification of 40x. This produced an imaging resolution of 0.28 µm/pix, where the histology image resolution is >10

times that of PLI. Due to the large slide size, many of the central sections were digitised in two images (labelled image 'a' and 'b').

Structure tensor analysis[36–39] was applied to the digitised Gallyas images to extract the primary fibre orientation per microscopy pixel (Fig. 4). Across a local neighbourhood of $150 \times 150$ pixels, the fibre orientations were then combined into a frequency histogram to produce a fibre orientation distribution for a ~ $40 \times 40\,\mu m$ 'superpixel'. Summary statistics were also extracted at the level of the superpixel, where the superpixel parameters include:

1. The fibre orientation distribution: orientations within the $40\,\mu m$ superpixel were combined into a frequency histogram (bin size = 2°).
2. The circular mean of the fibre orientation distribution.
3. The fibre orientation dispersion index at ~ $40\,\mu m$: a Bingham distribution was fitted to the fibre orientations within the superpixel and the dispersion parameter $\kappa$ was converted to the orientation dispersion index, $ODI = 2/\pi \, atan(1/\kappa)$.
4. The mean RGB value over the superpixel.

Unfortunately many of posterior Gallyas silver sections exhibit a tissue processing artefact resulting in inconsistent or patchy staining (Supplementary Fig. 12). This artefact is only observed in the posterior not anterior sections, and may be related to the formation of ice crystals during tissue processing. Remarkably, structure tensor analysis of slides with the staining artefact show smoothly varying orientations across the white matter that follow our neuroanatomical expectations (6b-e). Comparing structure tensor analysis of two adjacent slides, one without the artefact (6f) and one with the staining artefact (6g), we observe similar orientations, though with the artefactual slide showing reduced contrast in the grey matter (inset). As neither the PLI data nor the Nissl slides were affected by the same artefact, orientational information from either PLI or structure tensor analysis of the Nissl stained slides may be more reliable in these regions. One of the unassigned sets of tissue sections (currently in formalin), will likely be used to repeat the Gallyas staining to obtain myelin-density estimates across the posterior brain.

One primary limitation of structure tensor analysis is that it requires the user to specify a Gaussian smoothing kernel over which the intensity gradients are calculated. This study utilised a Gaussian kernel with sigma equivalent to 10 pixels, i.e. ~ $2.8\,\mu m$. Future work could consider the impact of kernels of different sizes.

### Co-registration of MRI and microscopy data

The polarised light images and structure tensor output were registered to the postmortem structural MR (MGE) image using TIRL (Fig. 5). The resolution of the images were 4, 40 and 300 μm respectively. The structural MR image was chosen as the target image as (i) it was the MR data with the highest spatial resolution and (ii) it provides good grey/white matter contrast. To drive the registration, we selected the microscopy images with white/grey matter contrast most similar to the structural MR image and with the most well-defined tissue boundaries. Consequently, for the Gallyas slides we used the structure tensor RGB 'thumb' image in CIELAB or L*a*b space. The L*a*b space is based on the opponent colour model of human vision, where any given colour is represented as the combination of lightness ('L'), a position along a red-green axis ('a') and that along a blue-yellow axis ('b'). The 'b' image was used to drive the MRI-microscopy registration because is shows fairly well defined tissue boundaries, that were difficult to determine in the RGB space. For PLI we used the 'inclination' map which, when compared to the transmittance images, are relatively unaffected by the 'bubble' artefact (Supplementary Fig. 11).

Co-registration of the BigMac microscopy data to the structural MRI required 2D to 3D registration without block face photos. This is equivalent to a TIRL *slice-to-volume* transform, as described in[19] (see Section 2.6 of[19] for more details). In brief, this transform is defined as a

chain of elementary 2D and 3D operations: a 2D scaling, rotation and translation, a 3D embedding, a 3D displacement field, a 3D rotation and translation, and a 3D affine matrix. Initial values and ranges for the transformation parameters were defined in a configuration file that we fine-tuned for the BigMac dataset in a trial and error process ("the optimised TIRL protocol"). The transformation parameters were then optimised in predefined combinations in an automated three-stage process. (1) We first provided approximate coordinates for the centre of the microscopy image in the MRI volume. Along with some tolerance of error, this defined a "slab" of the structural MRI within which the registration was optimised. The microscopy images were first resampled to the resolution of the structural MR data and then registered into the 3D imaging volume. The registration started with a rigid search to find a 3D surface in MRI space that best represented the "cutting plane". (2) A 3D affine matrix was optimised to account for shears. (3) Finally we accounted for non-linear deformations within the microscopy plane. For computational efficiency, the position of 32 automatically defined control points (distributed evenly across the slide) were optimised and the local displacement between these points was calculated by interpolation using Gaussian radial basis functions. The modality independent neighbourhood descriptor (MIND) cost function[42] was minimised during each part of the registration. Manually defined binary masks were used in all three stages of the registration to exclude cost contributions from background areas in the microscopy images.

After the registration was complete, the outputs of part 2 (only linear transforms) and 3 (with non-linear transforms) were qualitatively compared to the microscopy image and the output for which the tissue boundaries were most similar selected as the "user defined optimum".

The optimised TIRL protocol was found to produce good results across microscopy slides (see Supplementary Fig. 5 for example outputs). However, some users may wish to run their own registration (e.g. to register sections of the cerebellum for which transforms are not yet provided), or optimise the registration further for a specific slide or over a small region of interest. Instructions on how to achieve this, alongside example configuration files, and a script to easily assess the accuracy of the registration for any microscopy slide of interest, are provided in the online documentation and tutorials (c.f. Code Availability).

TIRL outputs a series of transformations which allow the user to transform either pixel or voxel coordinates, or orientational vectors between domains. Further scripts are provided to demonstrate how users can precisely map the high-resolution microscopy information to into the MRI volume, or vice versa.

The most anterior and posterior microscopy sections do not sample the corpus callosum meaning that there is no tissue directly connecting the two hemispheres. Once sectioned, the tissue from each brain hemisphere is fully disconnected and the distance between the hemispheres when mounted onto the slides is not meaningful. Therefore, each hemisphere was registered separately to the MR data with the aid of hand-drawn tissue masks. Similarly, the cerebrum was masked and registered separately to the cerebellum (ongoing work).

### Comparing fibre orientation distributions from microscopy and MRI

For a qualitative comparison of fibre orientations from coregistered MRI and PLI (Fig. 6a), postmortem diffusion MRI data ($b = 4\,ms/\mu m^2$, 250 gradient directions, 1 mm isotropic) were processed using the diffusion tensor model (FSL's dtifit,[43]) to produce maps of fractional anisotropy, FA, and the primary eigenvector, V1. These maps were warped to PLI space using TIRL[19], and V1 was projected onto the microscopy plane for comparison with PLI.

Fibre orientation distributions from MRI and microscopy were then compared on a voxelwise basis (Fig. 6b). Postmortem diffusion

MRI data ($b = 10 \text{ ms/μm}^2$, 1000 gradient directions, 1 mm isotropic) were processed using the Ball and Rackets model (BAR)[44], to estimate a single, disperse fibre orientation distribution (FOD) per voxel. Following a previously published method[4], the FOD was then projected onto the microscopy plane for direct comparison with those from PLI and histology. The microscopy FODs were created by first warping the microscopy orientations to MR space. The fibre orientations (from PLI or structure tensor analysis of the Gallyas-stained slides) within each MR voxel were then combined into a frequency histogram with respect to orientation angle (resolution = 2°), the output of which is shown in Fig. 6b.

Finally, the orientation dispersion index (ODI)[45] of each 2D FOD (from BAR, PLI and histology) was calculated according to ref. 4. An ODI of 0 indicates no dispersion, whilst an ODI of 1 describes isotropic dispersion. The ODI values were compared on to DTI FA[43] and microscopic FA from the simultaneous analysis of linear and spherical tensor encoded postmortem data ($b = 4, 7 \& 10 \text{ ms/μm}^2$, 1 mm isotropic)[47,48,69,70].

The analysis for Fig. 6c, d was repeated with in vivo MRI data. Session 1 data ($b = 1 \text{ ms/μm}^2$, 81 unique gradient directions, 1 mm isotropic) were similarly processed using the Ball and Rackets[44] and diffusion tensor models[43], and estimates of BAR ODI and DTI FA were compared to microscopy ODI estimated in the in vivo MR space.

For each scatter plot, the correlation coefficient $r$ was calculated using MATLABs[73] *fitlm* function, and the *p*-value calculated using an F-test comparing the regression model to a degenerate model with only a constant term. We analysed 3728 voxels in the white matter postmortem and 2915 voxels in vivo. The centrum semiovale mask contained 78 voxels.

### Hybrid diffusion MRI-microscopy 3D fibre reconstruction and tractography
For the hybrid orientations in Fig. 7, PLI data informed on fibre orientations within the microscopy plane, whilst the diffusion data provided through plane information. This facilitated reconstruction of 3D hybrid orientations at spatial resolution of the PLI data.

Postmortem diffusion MRI data ($b = 10 \text{ ms/μm}^2$, 1000 gradient directions, 1 mm isotropic) were analysed using the Ball and Stick (BAS) model to estimate 3 fibre populations per voxel, with 50 orientation estimates or 'samples' per population[20,21,74]. The PLI images were co-registered to the diffusion MRI data using an optimised TIRL protocol[19]. The in-plane angle was warped into the diffusion space[19,41] and compared to the BAS samples within the corresponding diffusion MRI voxel. To facilitate fair comparison, the BAS samples were projected onto the PLI plane. Samples from BAS fibre populations with signal fractions <0.05 were excluded. Finally, the PLI through-plane angle was approximated by that from the most similar BAS sample. This produced a hybrid diffusion MRI-PLI 3D fibre orientation per microscopy pixel.

The hybrid fibre orientations were then combined into 3D fibre orientation distributions (FODs). Here, a set of voxels were defined in diffusion space. In each voxel, the hybrid MRI-PLI fibre orientations populated a 3D 'orientation histogram' defined by 256 points evenly spaced across the sphere. Spherical harmonics of order 8 were then fitted to the normalised histogram. In spherical harmonic format, the hybrid diffusion MRI-PLI FODs could then be visualised in standard MRI viewers[60,75] and input into existing tractography methods[60]. Anatomically constrained streamline tractography was then performed using MRtrix (iFOD2)[60,76] with anatomical masks adapted from XTRACT[25].

### Reporting summary
Further information on research design is available in the Nature Portfolio Reporting Summary linked to this article.

## Data availability
The BigMac data, including minimally preprocessed data, are openly available via the Digital Brain Bank[6]: https://open.win.ox.ac.uk/DigitalBrainBank/#/datasets/anatomist. Example data can be accessed via the online viewer and the full dataset is available via a data sharing agreement to ensure the data is used for purposes which satisfy research ethics and funding requirements. Further documentation, including example images of all modalities, is available via https://open.win.ox.ac.uk/pages/amyh/bigmacdocumentation. Since the full dataset requires a considerable amount of memory (-1.8 TB), users may wish to request only a subset of relevant data. To facilitate this, the documentation includes an extensive file tree, listing available files as well as approximate memory requirements for different parts of the data. Additional data linking behaviour to fMRI (z-statistics maps) can be accessed via[17,18] and similar in vivo MRI (structural and resting-state fMRI) are available for another 19 subjects for cross-subject comparisons via the primate data exchange (PRIME-DE): https://fcon\_1000.projects.nitrc.org/indi/PRIME/oxford.html. Source data are provided with this paper.

## Code availability
Code is available via https://git.fmrib.ox.ac.uk/amyh/bigmacanalysis[77]. This includes basic MRI-microscopy tutorials as well as scripts related to data preprocessing, or reproducing the analyses presented.

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

## Acknowledgements

The authors would like to thank Katherine Bryant, Jesper Anderson, Frederik Lange, Moises Hernandez Fernandez, Paul McCarthy, Michiel Kleinnijenhuis and Menuka Pallebage Gamarallage for their advice at stages throughout this project. AFDH and IN were supported by the EPSRC, MRC and Wellcome Trust (grants EP/L016052/1 and WT202788/Z/16/A). IN was also supported by the Clarendon Fund in partnership with the Chadwyck-Healey Charitable Trust at Kellogg College, Oxford. AK and NS were supported by Cancer Research UK (grant number C5255/A15935). RM was supported by the BBSRC (grant BB/N019814/1). JS was funded by ANR grant (ANR-21-CE37-0016). KLM and SJ were supported by the Wellcome Trust (grants WT202788/Z/16/A, WT215573/Z/19/Z and WT221933/Z/20/Z). The Wellcome Centre for Integrative Neuroimaging is supported by core funding from the Wellcome Trust (WT203139/Z/16/Z). This research was funded in whole, or in part, by the Wellcome Trust (WT202788/16/A, WT215573/Z/19/Z). For the purpose of open access, the author has applied a CC BY public copyright licence to any Author Accepted Manuscript version arising from this submission.

## Author contributions

All authors reviewed the manuscript. AFDH: project lead; study conceptualisation; involved with acquisition of all postmortem MRI and microscopy data; developed acquisition protocols and processing pipelines; curated all data; conceived of and performed analyses; wrote and edited manuscript. INH: developed the TIRL software and protocol for co-registration of the MRI-microscopy data. AS: acquired and processed the microscopy data, including co-registration to MRI with hand drawn masks. MC: developed protocol for multiple diffusion tensor encoding. GD: processed tissue for microscopy, including sectioning, staining and mounting. TH: developed software for the Digital Brain Bank platform for downloading and viewing the data. AAK: optimised scanning protocols and acquired postmortem MRI. RBM: study conceptualisation; provided high resolution postmortem MRI protocol; advised on data processing and analysis. JM: developed software and protocols for polarised light imaging and structure tensor analysis. CS: facilitated slide scanning. NRS: facilitated postmortem MR scanning. JS: study conceptualisation; acquired in vivo data; provided postmortem brain & high resolution MRI protocol; facilitated MR acquisition and tissue processing for microscopy. SJ: study conceptualisation; advised on all acquisition protocols, processing pipelines and analyses; edited manuscript; provided resources and supervision. KLM: study conceptualisation; advised on all acquisition protocols, processing pipelines and analyses; edited manuscript; provided resources, supervision and funding.

## Competing interests

The authors declare no competing interests.
