## [Peer Review File · Nature Communications]

An open resource combining multi-contrast MRI and
microscopy in the macaque brainReviewer #1 (Remarks to the Author):

Thank you for the opportunity to review this wonderful article. The BigMac dataset is an incredible resource, important for methodological validation and foundational neuroscience. In this article, the authors present the outcomes of an elaborate, well-designed diffusion imaging protocol that is coupled with high-quality microscopy.

I was greatly pleased that the authors go beyond simply releasing the dataset and delve into several key questions. In particular, the authors (i) demonstrate the advantages of higher angular resolution for reconstruction of long-range tracks, (ii) challenge common assumptions on the source of birefringence in PLI, (iii) demonstrate the advantage of ultra-high-resolution histology, even compared to PLI and (iv) suggest a novel approach for approximating high resolution structural connectivity from hybrid MRI-microscopy.

Furthermore, the article was a pleasure to read. I would like to congratulate the authors on this outstanding piece of work.

I have a few minor comments/suggestions, that I hope the authors could address.

Methods

1. Could the authors provide a timeline that overviews the experiments? In particular, including age of each in vivo session, age at death, time to post-mortem scanning, time to fixation.
2. "As the diffusion properties of brain tissue are highly dependent on the tissue temperature [54], the temperature was controlled by passing air at the constant temperature of 20°C." I assume this is a mistype of 20 or 21°C?
3. More information on the TIRL models should be incorporated into the present manuscript to ensure full comprehension of the co-registration. For example, what is meant by "optimised TIRL protocol". For each application, the precise series of transformations should be detailed. Additionally, how does TIRL achieve the 2D to 3D registration?

Results

4. In Figure 3, please specify in "a)" which boxes are A-D.
5. The relative absence of myelin staining in the molecular layer of the cerebellum in combination with the strong PLI signal is an interesting observation. To aid reader from outside a physics background, could the authors elaborate on the effect of positive birefringence on the PLI signal. Specifically, I am curious what is meant by "would result in PLI orientations tangential to those observed". Is the angle of the tangent known? In other words, could one simulate/model the alternative hypotheses? It would be nice to point this out for future work.
6. Were any quantitative evaluations performed on the registration (especially for dMRI to PLI)? In particular, how does the procedure perform in areas of significant tissue deformation?

Conclusions

7. I would contend that it's not the 2D nature of the microscopy that prohibits 3D reconstruction, but rather the incompleteness of the 2D sections in representing the whole brain. If all sections were available in one modality, an entire 3D reconstruction could be achieved.

Reviewer #2 (Remarks to the Author):

In this manuscript, Howard et al. describe a new dataset, BigMac, combining in vivo MRI, ex vivo MRI, and multiple types of microscopy in a macaque brain. These types of

data are extremely valuable—the careful curation and within-subject nature mean that they can potentially be used to reconstruct connectomes in a more accurate fashion.

This is a somewhat unusual manuscript for Nature Communications, because it describes the release of a dataset rather than a particular discovery or advance. I thought carefully about what criteria should be used to evaluate such a study; I will comment on some of them below, but ultimately, I think this is a decision for editorial staff.

Major concerns

Included in the dataset are just data from a single animal, and it turns out this animal has bilateral lesions of the orbitofrontal cortex from prior experiments, as well as an abnormality in the left hemisphere. Both of these obviously make the dataset somewhat suboptimal. There is still a great deal that can be done with these data, but it seems like a troublesome point.

I did poke around the provided data, and one thing stuck out. In the Gallyas silver and PLI images, there are some pretty dramatic differences between the two hemispheres. It's noticeable in part because the corpus callosum is pink on one side, and has a strong yellow presence on the other. It can be seen also in many other bundles. I'm trying to figure out what would cause this effect. Obviously, the tissue could have been cut at a L-R slight angle (although this makes the dataset less than ideal), but I think the resulting differences would be more gradual than they are. I did not see such differences across the two hemispheres in the diffusion MRI data. I do wish there had been an easy way to quality check the registration across data types, but I think in the reviewer-provided files, each type was kept separate. The MRI-microscopy correlations shown in Figure 6 do appear somewhat low; I wonder whether this is related?

I am concerned about accessibility and user base. The authors say that the dataset will be available using a Materials Transfer Agreement (MTA). Some smaller portions of the data will be deposited in various places that are more accessible. In my experience, the use of an MTA will drastically limit how much a dataset is used by the public. It adds a substantial layer of bureaucracy that will differ institution-to-institution. It is possible that the authors have no control over this, and that is frustrating. Furthermore, the data are being released with some registration & analysis scripts, but not with any new toolboxes for working with the data. I'm trying to figure out what the 'audience' will be for the dataset, and I'm just concerned it will not be large.

Minor comments

In headings and figures, the different types of microscopy should be differentiated.

At the top of p.8, the descriptions of the SLF II need citations to prior work to cement the definition of the tract and the notion of the false positive.

On p. 10, this claim "Furthermore, in the deep white matter we see different tissue 'textures' as well as large scale, 'wave-like' undulations of fibres in the corpus callosum (D). Though this a region which is often considered coherently ordered with little fibre dispersion, these data corroborate previous observations of fibre dispersion or incoherence in the corpus callosum [4]" is very interesting, but I don't know what it means. Maybe another figure on these waves would be helpful? I can't tell what feature of the CC is being referenced.

The images of the Gallyas silver match what my concern has always been about using this stain to estimate fiber orientations and build connectomes. Where fibers are relatively sparse, it is possible to distinguish single axons. But where there are many fibers together, the fibers all blur together in a mass, and all sense of individual axons is lost. Does this sound right to the authors? If so, I think stating some of the weaknesses

of this approach (it may not be the be-all, end-all) would be helpful.

Reviewer #3 (Remarks to the Author):

Howard et al. present BigMac, a openly shared multimodal dataset of the macaque brain. This dataset combines in vivo MRI, ex vivo MRI and multiple histological stains. The focus of the author's work here is cross-comparison of diffusion-derived measures of brain anatomy. The high resolution diffusion data, coupled with PLI and fibre orientation estimation from sections stained for myelin is fantastic resource to better understand cerebral fibre organisation and the microstructural underpinnings of DTI. The open sharing of this dataset is commendable will ensure maximum value from this dense resource. While the authors state that the presented dataset will be enriched with additional histological staining, the current data are sufficient to be a unique and valuable addition to multiscale neuroscience.

The MRI diffusion imaging, histological acquisition and preprocessing appear carefully thought out. The authors are clear with in describing the potential artefacts including orbitofrontal lesions, bubbles and mounting-related biases that were addressed during acquisition. The manuscript is primarily a description of the resource, describing the acquisition, highlighting interesting features from visual inspection of the data and some preliminary analyses including comparison of connectivity between 64-direction and 1000-direction diffusion images and comparison of diffusion-microstructural estimates between MRI and histology.

My main comments relate to these statistical analyses. While I acknowledge they are intended as demonstrations of data quality and the unique analyses afforded by Big Mac, I think these should be strengthened.

1. Figure 2. Comparison of 64 vs 1000 direction DTI. The authors find a 50-150 fold increase in connectivity between lateral homotopic regions. This appears to be an extremely large change, with 3 areas exhibiting orders of magnitude greater numbers of fibres and needs some additional context. Are fibres for these 3 regions expected to be underestimated by 64 direction scans (i.e. some previously recognised issue in DTI)? Do these new values reflect something closer to ground truth using 1000 directions? Is there a way the authors could propose a mechanism as to why these measures are so discordant? If possible, it would also be powerful to demonstrate the number of connections at 1000 are closer to some external "ground truth" estimate (e.g. tracer based estimates of connectivity strength)? While I appreciate that streamline length relationship could be playing a role but the huge jump in streamlines shown in 2c panel 5 is beyond that predicted with 2c panel 6, which shows a steadily length-dependent recovery in streamlines.
2. Figure 6. The plots comparing MRI and histological estimates (fig 6c) would be strengthened by more data points from other sections and some extra statistical analysis. It seems odd given the richness of the dataset that you are restricting these analyses to a few voxels from a single section. Why not increase the sample size for these analyses to include samples from multiple aligned sections across the brain?
3. It appears as though some of these results are heavily driven by individual outliers e.g. BAR ODI vs PLI ODI, where a single data point seems to be strongly influencing this line of best fit. Multiple sections would help the impact of one noisy voxel, and statistical significance testing and confidence intervals on the fitted relationships would also be informative.
4. Could it be made clearer which diffusion MRI is data is being compared throughout the manuscript and particularly in this analysis? I think it is all from the post mortem data, but as the authors collected in vivo also it would be useful to clarify which acquisition used (e.g. in the fig 6 caption).
5. Is there a reason why both ex vivo and in vivo DTI weren't included in the fig 6

analyses? This would show the full range of scales and modalities in the resource

6. It appears as though the PLI data is underestimating ODI relative to BAR and histology (Fig 6c). BAR and histology have a roughly 1:1 relationship, while the PLI ODI estimates are almost always above the $y = x$ line. Do you have an explanation as to why this is the case? Perhaps this pattern might disappear with more extensive data sampling or maybe there is a systematic bias in one of the measures.

7. The authors have described cerebellar masking as "ongoing work". Could you clarify to what extent the cerebellum has been co-registered to the structural MRI and whether researchers can already carry out joint analyses of these brain areas with this release?

Thanks for sharing sample data for this review. It's not clear where the code is going to be made available – is there a code repository that I've missed?

I spotted some minor typos / clarifications:

Line 28 Scientific communitys >community's

Line 44 consider rewording: These analyses require complex computational signal modelling with many strong assumptions.

Do you mean the assumptions are strong, or that the modelling is strongly dependent on assumptions?

Line 57 has co-registered MRI and microscopy data for to facilitate meaningful voxelwise comparisons

383 "corss" > cross

449 – Is 201 degrees air temperature correct? That sounds hot, like a temperature at which the tissue (and researchers) might begin to cook.

Response to reviewers

Many thanks to the reviewers for their careful consideration of the manuscript and kind words regarding our work. Below we address each of their comments in turn, where related text lifted from the manuscript is shown in *italics*, and edited text is shown in red. Further, we provide both a clean and mark-up version of the revised manuscript.

Please note, code to recreate the analyses performed in the paper are openly available via <https://git.fmrib.ox.ac.uk/amyh/bigmacanalysis>. At the current time, we do not provide open code to recreate the outputs in Figure 7 (hybrid tractography) as this is the subject of active, ongoing work within the group. However, this code is made privately available to reviewers (HybridOrientations.zip password: Howard). We are very happy to again provide example copies of the BigMac data to reviewers on request.

Reviewer 1

Thank you for the opportunity to review this wonderful article. The BigMac dataset is an incredible resource, important for methodological validation and foundational neuroscience. In this article, the authors present the outcomes of an elaborate, well-designed diffusion imaging protocol that is coupled with high-quality microscopy.

I was greatly pleased that the authors go beyond simply releasing the dataset and delve into several key questions. In particular, the authors (i) demonstrate the advantages of higher angular resolution for reconstruction of long-range tracks, (ii) challenge common assumptions on the source of birefringence in PLI, (iii) demonstrate the advantage of ultra-high-resolution histology, even compared to PLI and (iv) suggest a novel approach for approximating high resolution structural connectivity from hybrid MRI-microscopy.

Furthermore, the article was a pleasure to read. I would like to congratulate the authors on this outstanding piece of work.

Many thanks for your very kind words about this work.

I have a few minor comments/suggestions, that I hope the authors could address.

Methods

1. Could the authors provide a timeline that overviews the experiments? In particular, including age of each in vivo session, age at death, time to post-mortem scanning, time to fixation.

The following information has now been included as a timeline in section 6.1: Data acquisition:

The acquisition timeline for presented data was as follows:

- *In vivo MRI session 1: April 2010 (4 years old)*
- *In vivo MRI session 2: January 2017 (10 years 9 months old)*
- *Perfusion fixation at death: December 2017 (11 years 7 months old)*
- *Postmortem MRI: March-April 2018 (3-4 months postmortem)*
- *Postmortem microscopy: 2018 onwards*

Note that the time to fixation was essentially zero as the bigmac brain was perfusion fixed at the time of sacrifice.

2. As the diffusion properties of brain tissue are highly dependent on the tissue temperature [54], the temperature was controlled by passing air at the constant temperature of 20°C. I assume this is a mistype of 20 or 21°C?

Many thanks for catching this typo. It has now been corrected.

As the diffusion properties of brain tissue are highly dependent on the tissue temperature [1], the temperature was controlled by passing air at the constant temperature of 20° C.

3. More information on the TIRL models should be incorporated into the present manuscript to ensure full comprehension of the co-registration. For example, what is meant by optimised TIRL protocol. For each application, the precise series of transformations should be detailed. Additionally, how does TIRL achieve the 2D to 3D registration?

TIRL is a very flexible framework that includes a series of modular transforms or “stages” that can be combined together depending on the task in hand. A detailed description of the full TIRL functionality has recently been published in Huszar et al. 2023: <https://doi.org/10.1016/j.neuroimage.2022.119792> [2]. Co-registration of the BigMac data without block face photos utilised the TIRL slice-to-volume transform (“Stage 3” in the above manuscript). Additional details of its application to the BigMac dataset have been included in Methods Section 6.5 *Coregistration of MRI and microscopy data*:

Co-registration of the BigMac microscopy data to the structural MRI required 2D to 3D registration without block face photos. This is equivalent to a TIRL slice-to-volume transform, as described in [2] (see Section 2.6 of [2] for more details). In brief, this transform is defined as a chain of elementary 2D and 3D operations: a 2D scaling, rotation and translation, a 3D embedding, a 3D displacement field, a 3D rotation and translation, and a 3D affine matrix. Initial values and ranges for the transformation parameters were defined in a configuration file that we fine-tuned for the BigMac dataset in a trial and error process (“the optimised TIRL protocol”). The transformation parameters were then optimised in predefined combinations in an automated three-stage process. 1) We first provided approximate coordinates for the centre of the microscopy image in the MRI volume. Along with some tolerance of error, this defined a “slab” of the structural MRI within which the registration was optimised. The microscopy images were first resampled to the resolution of the structural MR data and then registered into the 3D imaging volume. The registration started with a rigid search to find a 3D surface in MRI space that best represented the “cutting plane”. 2) A 3D affine matrix was optimised to account for shears. 3) Finally we accounted for non-linear deformations within the microscopy plane. For computational efficiency, the position of 32 automatically defined control points (distributed evenly across the slide) were optimised and the local displacement between these points was calculated by interpolation using Gaussian radial basis functions. The modality independent neighbourhood descriptor (MIND) cost function [3] was minimised during each part of the registration. Manually defined binary masks were used in all three stages of the registration to exclude cost contributions from background areas in the microscopy images.

After the registration was complete, the outputs of part 2 (only linear transforms) and 3 (with non-linear transforms) were qualitatively compared to the microscopy image and the output for which the tissue boundaries were most similar selected as the “user defined optimum”.

The optimised TIRL protocol was found to produce good results across microscopy slides (see Supplementary Figure 5 for example outputs). However, some users may wish to run their own registration (e.g. to register sections of the cerebellum for which transforms are not yet provided), or optimise the registration further for a specific slide or over a small region of interest. Instructions on how to achieve this, alongside example configuration files, and a script to easily assess the accuracy of the registration for any microscopy slide of interest, are provided in the online documentation and tutorials.

Results

4. In Figure 3, please specify in a) which boxes are A-D.

Apologies this was unclear. Box A comes from image 8 on Figure 3a left, B from image 3, C from image 4 and D from image 6. The following text has been added to the caption:

a) Example PLI throughout the brain. Image 1 = most anterior, 8 = most posterior of the images shown. The myeloarchitecture is viewed in great detail due to the $4\ \mu\text{m}$ PLI resolution. We observe fibres projecting into the cortex (right hand panel part A), into and through subcortical structures (B), around the pons (C) and across the cerebellum (D). Inset A comes from image 8, B from image 3, C from image 4 and D from image 6. In D, the blue arrows point to the gyri crown.

5. The relative absence of myelin staining in the molecular layer of the cerebellum in combination with the strong PLI signal is an interesting observation. To aid reader from outside a physics background, could the authors elaborate on the effect of positive birefringence on the PLI signal. Specifically, I am curious what is meant by would result in PLI orientations tangential to those observed. Is the angle of the tangent known? In other words, could one simulate/model the alternative hypotheses? It would be nice to point this out for future work.

Our PLI analysis outputs orientations that describe the fast axis of the birefringent medium i.e. the axis along which the refractive index is lower such that light travels faster. How this maps onto the orientation of the fibre depends on whether the source birefringence is positive or negative uniaxial with respect to the longitudinal or radial axis of the fibre.

The biological basis of axon birefringence is typically attributed to two effects. First, axon organelles (i.e. the microtubules and neurofilaments) and myelin membrane proteins result in positive uniaxial birefringence with respect to the longitudinal axis of the fibre. This implies that the mediums fast axis (output from our PLI analysis) is perpendicular to the longitudinal axis of the fibre. Second, the anisotropic structure of the myelin lipid bilayer results in a microscopic positive birefringence with respect to the radial axis of the fibre. At the macroscopic level (i.e when imaging multiple fibres through the tissue plane), this results in uniaxial negative birefringence [4] for optical resolutions larger than the fibre radius - an assumption which generally holds in most practical PLI setups. This results in the fast axis (estimated from PLI) being parallel to the longitudinal axis of the fibre (our typical model).

It is generally assumed that in myelinated fibres, the birefringence associated with the lipid bilayer dominates (i.e. the macroscopic model of uniaxial negative birefringence with respect to the longitudinal axis of the fibre). Hence, we assume the orientations output from PLI indicate the orientation of underlying fibres within the white matter. However, some previous work suggests that for low diameter (and thus typically low/un-myelinated) fibres, the birefringence associated with organelles of myelin proteins can dominate. This would indicate underlying fibres oriented orthogonal to our PLI outputs.

This leads to the juxtaposition observed in our and others data [5]. For the observed PLI signal in the cerebellar molecular layer to be associated with parallel fibres (ie. that the PLI outputs are aligned with the underlying fibres), we would typically attribute the source birefringence to the myelin lipid bilayer. However, our histology suggests that these fibres have little or no myelin. Here we hypothesise that there could be two effects: i) that the PLI signal is sensitive to not only myelin, but also other aspects of microstructure orientational coherence [6], or ii) that the parallel fibres have some small amount of myelin which is detected by PLI but not picked up by the histology stain. However these speculations require experimental justification and further investigation beyond the scope of this study. Advanced simulation of the PLI signal for fibres with little or no myelin may help provide insight, for example using the fastPLI simulation toolbox by Matuchke, Amunts and Axer [7].

Though a detailed explanation is beyond the scope of this paper, the following text has been included in Section 2.3.1 *Polarised light imaging*

Although PLI signal from brain tissue is typically associated with the myelin, histological data did not support the presence of myelinated fibres in the molecular layer (Figure 3b left). Furthermore, non-myelinated fibres often exhibit positive birefringence with respect to the longitudinal axis of the fibre due to the presence axon organelles and myelin membrane proteins, rather than negative birefringence associated with the lipid bilayer

Manscript Figure 3: Polarised light imaging in BigMac. The colours correspond to orientations described by the 2D colour wheel in image D. Note, this is different from the standard colour representation in diffusion MRI. a) Example PLI throughout the brain. Image 1 = most anterior, 8 = most posterior of the images shown. The myeloarchitecture is viewed in great detail due to the 4 μm PLI resolution. We observe fibres projecting into the cortex (right hand panel part A), into and through subcortical structures (B), around the pons (C) and across the cerebellum (D). Inset A comes from image 8, B from image 3, C from image 4 and D from image 6. In D, the blue arrows point to the gyri crown. b) We hypothesise that the PLI signal in the cerebellar molecular layer can be attributed to parallel fibres. b, left) The yellow and green arrows point to the granular layer and the molecular layer respectively. b, right) The structure of the cerebellar cortex. This highly simplified schematic focuses solely on the granule and Purkinje cells, to illustrate the parallel fibres in the molecular layer. Note, the dendritic tree of the Purkinje cells has highly anisotropic dispersion. Here we see the axis of least dispersion, where the Purkinje dendritic tree fans out most in the through-page orientation.

[8, 4]. Such positive birefringence would result in PLI orientations *orthogonal* to those observed [4], as the “fast axis” of the medium output from our PLI analysis would now indicate the radial rather than longitudinal axis of the fibre. Here there could be two effects: i) that the PLI signal is sensitive to not only myelin, but also other aspects of microstructure orientational coherence [6], or ii) that the parallel fibres have some small amount of myelin which is detected by PLI but not picked up by the histology stain. ... Future work is required to fully understand the origin of this birefringence in the molecular layer, though Figure 3 provides some evidence for PLI sensitivity to coherently orientated, anisotropic structures irrespective of their degree of myelination [5, 9], that may be omitted from classic myelin histology. *Simulations of the PLI signal, for example based on the fastPLI framework [7], may help provide insight.*

6. Were any quantitative evaluations performed on the registration (especially for dMRI to PLI)? In particular, how does the procedure perform in areas of significant tissue deformation?

Quantitative evaluation of the registration is challenging due to difficulties in generating a ground truth that can be compared between the MRI and microscopy data which often show considerably different contrasts (i.e. equivalent grey-white boundaries can be difficult to define, and tissue edges are often challenging to define in the microscopy images due to lack of contrast). The accuracy of such quantitative evaluations is thus very reliant on the accuracy of the definition of landmarks or segmentations within the native MRI and microscopy images, which is often worse than the accuracy of the registration algorithm. In Huszar et al. [2], we recently evaluated TIRLs performance using simulated data (with a known ground truth), where the slice-to-volume algorithm was shown to produce sub-voxel level accuracy (0.13 mm). These simulations included more extreme deformations than those generally observed within the BigMac sections.

We appreciate however that it is important for users to be able to easily verify the registration quality for any slide or region of interest. Consequently, we have now provided a simple Jupyter notebook on the BigMac GitLab page that allows the user to assess the quality of co-registration for any microscopy section of interest (https://git.fmrib.ox.ac.uk/amyh/bigmacanalysis/-/blob/main/TIRL/TIRL_checking_registration.ipynb). With the notebook, the user can map any MR image onto a microscopy slide of interest. This can then be qualitatively compared with the microscopy image, either via mosaic images similar to those provided in the Figure below (Supplementary Figure 5), or using the overlay functions in an image viewer such as fsleyes (recommended). This solution is especially beneficial for users interested in a particular region of interest, such as the hippocampus, as they are able to easily assess the registration quality within this specific region.

The following text has been added to Section 6.5 *Co-registration of MRI and microscopy data*:

The optimised TIRL protocol was found to produce good results across microscopy slides (see Supplementary Figure 5 for example outputs). However, some users may wish to run their own registration (e.g. to register sections of the cerebellum for which transforms are not yet provided), or optimise the registration further for a specific slide or over a small region of interest. Instructions on how to achieve this, alongside example configuration files, and a script to easily assess the accuracy of the registration for any microscopy slide of interest, are provided in the online documentation and tutorials.

For transparency, example outputs demonstrating the registration quality for every 20th section are now also included in Supplementary Figure 5. This includes outputs from mapping between either PLI or Gallyas to both postmortem structural MRI (MGE, 0.3 mm isotropic) and postmortem diffusion MRI ($b = 10 \text{ ms}/\mu\text{m}^2$, 1 mm isotropic). We highlight example regions where the coregistration is less successful with yellow boxes.

Conclusions 7. I would contend that its not the 2D nature of the microscopy that prohibits 3D reconstruction, but rather the incompleteness of the 2D sections in representing the whole brain. If all sections were available in one modality, an entire 3D reconstruction could be achieved.

It is a good point that 3D reconstruction of the microscopy data at mesoscopic resolution can be achieved if consecutive sections are acquired with a single microscopy contrast, such as the BigBrain 3D reconstruction at 20 microns [10]. The original text intended to refer to the microscopy not being able to visualise the 3D

<--- Anterior ----- Postmortem structural MRI (MGE) mapped to PLI slides ----- Posterior -

<--- Anterior ----- Postmortem diffusion MRI (b=10, 1mm) mapped to PLI slides ----- Posterior -

.....
<--- Anterior ----- Postmortem structural MRI (MGE) mapped to Gallyas histology slides ----- Posterior -

<--- Anterior ----- Postmortem diffusion MRI (b=10, 1mm) mapped to Gallyas histology slides ----- Posterior -

Supplementary Figure 5: Example registration outputs when mapping either the postmortem structural MRI (MGE, 0.3 mm isotropic) or postmortem diffusion MRI ($b = 10 \text{ ms}/\mu\text{m}^2$, 1 mm isotropic) data onto the microscopy plane. Each box in the mosaic image alternates between showing the co-registered MRI output (lower resolution, black background at tissue edges) or the native microscopy image (higher resolution, white background at tissue edge). 1 in 20 slides are shown for PLI (top, inclination image) and Gallyas histology (bottom, RGB structure tensor output converted to greyscale). Note the diffusion images are blurred due to interpolation. The yellow boxes indicate regions in which the registration is less accurate. In the interest of space, only the right hemisphere is shown.

microstructure within each microscopy section (e.g. the 3D visualisation of single cells, or 3D orientations of nerve fibres), which could be imaged with more advanced 3D microscopy methods or z-stack imaging. To make this clearer, and to add the reviewers point about single modality whole brain 3D reconstruction, the text has subsequently been edited to:

The dataset has several limitations including: the 2D nature of the microscopy, which precludes 3D visualisation of the cell bodies or nerve fibres within each microscopy section; the multimodal nature of the microscopy prohibits whole-brain 3D reconstruction of the tissue microstructure from a single contrast at high resolution (as in e.g. the BigBrain dataset [10]);...

Reviewer 2

In this manuscript, Howard et al. describe a new dataset, BigMac, combining in vivo MRI, ex vivo MRI, and multiple types of microscopy in a macaque brain. These types of data are extremely valuable; the careful curation and within-subject nature mean that they can potentially be used to reconstruct connectomes in a more accurate fashion.

This is a somewhat unusual manuscript for Nature Communications, because it describes the release of a dataset rather than a particular discovery or advance. I thought carefully about what criteria should be used to evaluate such a study; I will comment on some of them below, but ultimately, I think this is a decision for editorial staff.

Major concerns

1. Included in the dataset are just data from a single animal, and it turns out this animal has bilateral lesions of the orbitofrontal cortex from prior experiments, as well as an abnormality in the left hemisphere. Both of these obviously make the dataset somewhat suboptimal. There is still a great deal that can be done with these data, but it seems like a troublesome point.

Many thanks for your comment and for taking the time to review this manuscript. We agree there are limitations in any resource based on a single subject, and are already developing plans to acquire data on a second BigMac brain. However, the deep phenotyping of each single brain is very time intensive, often taking multiple years to acquire. Data from the second BigMac brain will be made openly available in due course. Further, we believe it is ethically important to get as much scientific knowledge as possible out of a research animal, particularly primates. Thus, while we acknowledge the limitations created by the presence of lesions, we feel that by acquiring postmortem data in an animal that was studied for a very different purpose in life presents considerable added value along the lines of the 3R principles (replacement, reduction, refinement).

Nonetheless, there are still numerous opportunities for the data presented. Firstly, there is not existing data linking the range of multi-contrast MRI and microscopy as included here (in vivo MRI, extensive diffusion MRI, PLI and histology). Understanding the fundamental mappings between MRI and the underlying microstructure requires voxelwise analyses for which BigMac provides extensive data. Here, having examples of more “atypical” microstructure, such as that surrounding the bilateral lesions where we expect some related microstructural changes, may be of benefit as it provides the opportunity to investigate whether MRI metrics have sensitivity to subtle pathological changes, which is particularly relevant in the clinical setting. For those interested in more neuroanatomical investigations, the lesions/abnormality affect only a small portion of the tissue, meaning that the majority of the brain can be studied with relatively minimal impact.

Furthermore, we do not believe the BigMac dataset should be viewed in isolation, but can be utilised in combination with the wealth of existing and often invasive data from other macaques. This includes in vivo data and postmortem MRI from other animals (e.g. in the PRIME-DE database), as well as large number of existing microscopy, both from our own lab and others. These data allow us to test between-

subject variability within smaller regions of interest, or fewer modalities. As the validation of results in other subjects often does not require the richness of data in discovery, additional animals with e.g. only a few microscopy slides or limited MRI, can be used to validate a result discovered in BigMac. For example, one of the strengths of the BigMac data is that it includes densely sampled microscopy data that be used to define e.g. cortical boundaries for multimodal segmentation of the cortex. Validation of a boundary created in BigMac could potentially be performed against another animal which has relatively few histology slides over some smaller region of interest.

Even before publication, we have already received 10 external requests for the BigMac data across multiple countries and groups, demonstrating its value as an open resource to the community for numerous applications.

That these data come from only a single subject with lesions and abnormalities has explicitly added in the "limitations" paragraph of Section 3. *Conclusion:*

The dataset has several limitations including: ... Finally, we acknowledge that this data was acquired from a single macaque brain, with lesions in the orbitofrontal cortex and an abnormality in the left hemisphere. This may limit the datasets suitability for specific applications.

Nonetheless, as a unique, multimodal resource that complements existing open data (for cross-modality, cross-subject or cross-species investigations, as well as data from invasive macaque studies not possible in humans), the BigMac dataset will enable neuroscientists to ask new and fundamental questions. For example, ...

2. I did poke around the provided data, and one thing stuck out. In the Gallyas silver and PLI images, there are some pretty dramatic differences between the two hemispheres. Its noticeable in part because the corpus callosum is pink on one side, and has a strong yellow presence on the other. It can be seen also in many other bundles. Im trying to figure out what would cause this effect. Obviously, the tissue could have been cut at a L-R slight angle (although this makes the dataset less than ideal), but I think the resulting differences would be more gradual than they are. I did not see such differences across the two hemispheres in the diffusion MRI data. I do wish there had been an easy way to quality check the registration across data types, but I think in the reviewer-provided files, each type was kept separate. The MRI-microscopy correlations shown in Figure 6 do appear somewhat low; I wonder whether this is related?

The different colours between the two hemispheres in the PLI and Gallyas structure tensor outputs is expected, as this data is represented by a 2D colour wheel (due to the 2D nature of the orientations), rather than the 3D colour scheme used in diffusion MRI. If we refer to the colour wheel in the microscopy figure below (Manuscript Figure 4), the pink part of the corpus callosum indicates orientations going approximately north east (if we convert the SI-RL axes of the image to the north-south, east-west axes of a compass), whilst the yellow orientations are going north west. This follows the underlying neuroanatomy, with fibres descending from the centrum semiovale, through the corpus callosum, and out to the other hemisphere. Figure 6a shows example DTI data projected onto the 2D plane and coloured using the 2D colour wheel. Both in Figure 6a and in Figure 4 below we see good agreement with the orientations from the PLI with the different datasets corroborating the information from one another.

We have ensured that each Figure using the 2D colour scheme includes the 2D colour wheel, and have added the following clarification to the caption of Figure 3 where this colour scheme is first introduced.

Figure 3: Polarised light imaging in BigMac. The colours correspond to orientations described by the 2D colour wheel in image D. Note, this is different from the standard 3D colour representation in diffusion MRI.

As mentioned to the reviewer above, we appreciate the value of users being able to easily check the quality of the co-registration as suggested. Consequently, we have now provided a simple jupyter notebook on the Big-

Manuscript Figure 4: Gallyas silver stained histology in BigMac. a) Example digitised slides. With a spatial resolution of $0.28 \mu\text{m} / \text{pixel}$, we see the myelinated fibres in great detail, visualising single axons at the grey/white matter boundary (A-F) and fibre undulations in the deep white matter (D). b) Structure tensor analysis was applied to the Gallyas silver stained slides to estimate a fibre orientation per $0.28 \mu\text{m} \text{ pixel}$. c,d) The fibre orientations derived from the Gallyas silver stained slide are compared to an adjacent PLI slide. We see remarkable consistency between the images with both modalities capturing the myeloarchitecture in detail. e) Both modalities show the detailed organisation of the hippocampus and surrounding white/grey matter (yellow box), as well as the corticospinal tract (white box).

Mac GitLab page that allows the user to assess the quality of co-registration for any microscopy section of interest: https://git.fmrib.ox.ac.uk/amyh/bigmacanalysis/-/blob/main/TIRL/TIRL_checking_registration.ipynb. This notebook maps structural or diffusion MRI to either PLI or Gallyas images, where other MR data can be easily mapped in a similar fashion. Example outputs demonstrating the registration quality for every 20th section are additionally now included in Supplementary Figure 5. Further, images where the structural MRI has been mapped onto the microscopy plane are now provided precomputed for each microscopy slide when users download the data. This provides a very quick way for users to examine the registration accuracy for each slide of interest.

The following text has been added to Section 6.5 *Co-registration of MRI and microscopy data*:

The optimised TIRL protocol was found to produce good results across microscopy slides (see Supplementary Figure 5 for example outputs). However, some users may wish to run their own registration (e.g. to register sections of the cerebellum for which transforms are not yet provided), or optimise the registration further for a specific slide or over a small region of interest. Instructions on how to achieve this, alongside example configuration files, and a script to easily assess the accuracy of the registration for any microscopy slide of interest, are provided in the online documentation and tutorials.

3. I am concerned about accessibility and user base. The authors say that the dataset will be available using a Materials Transfer Agreement (MTA). Some smaller portions of the data will be deposited in various places that are more accessible. In my experience, the use of an MTA will drastically limit how much a dataset is used by the public. It adds a substantial layer of bureaucracy that will differ institution-to-institution. It is possible that the authors have no control over this, and that is frustrating. Furthermore, the data are being released with some registration & analysis scripts, but not with any new toolboxes for working with the data. Im trying to figure out what the audience will be for the dataset, and Im just concerned it will not be large.

Due to our funders and ethics responsibilities, we are required to make the data available via an MTA rather than as a freely available download online [11]. We appreciate the reviewers concerns with respect to access, and so have recently carefully reviewed our process to make it as easy as possible for users on the other end. In brief, when someone requests access for the data, we ask them to fill in a short form outlining their name, the title and a short description of the proposed project (as is common e.g. in the HCP application), any collaborators, and the details of the University representative who can sign the MTA. We then send back a filled contract for signing. After the contract is signed, we provide the user access to an online OneDrive folder with the requested data from which they can download either the full dataset or a subset of data as desired.

Note, we have already transferred the BigMac data to several users with completed MTAs, suggesting that the MTA process is not overly prohibitive.

To date we have made multiple analysis scripts available on the BigMac GitLab page <https://git.fmrib.ox.ac.uk/amyh/bigmacanalysis/> to facilitate others to build upon the multiple analyses presented in the paper. Further, we provide a tutorial for MRI-microscopy co-registration, which has successfully been used by several early users who are utilising the BigMac data for their own projects. This online repository of tutorials and example analyses will continue to be built upon in the future.

The URLs for the online documentation and analysis tools are now explicitly provided in the manuscript in Section 5 *Code availability*:

Code is available via <https://git.fmrib.ox.ac.uk/amyh/bigmacanalysis/>. This includes basic MRI-microscopy tutorials as well as scripts related to data preprocessing, or reproducing the analyses presented.

Minor comments

<--- Anterior ----- Postmortem structural MRI (MGE) mapped to PLI slides ----- Posterior -

<--- Anterior ----- Postmortem diffusion MRI (b=10, 1mm) mapped to PLI slides ----- Posterior -

<--- Anterior ----- Postmortem structural MRI (MGE) mapped to Gallyas histology slides ----- Posterior -

<--- Anterior ----- Postmortem diffusion MRI (b=10, 1mm) mapped to Gallyas histology slides ----- Posterior -

Supplementary Figure 5: Example registration outputs when mapping either the postmortem structural MRI (MGE, 0.3 mm isotropic) or postmortem diffusion MRI ($b = 10 \text{ ms}/\mu\text{m}^2$, 1 mm isotropic) data onto the microscopy plane. Each box in the mosaic image alternates between showing the co-registered MRI output (lower resolution, black background at tissue edges) or the native microscopy image (higher resolution, white background at tissue edge). 1 in 20 slides are shown for PLI (top, inclination image) and Gallyas histology (bottom, RGB structure tensor output converted to greyscale). Note the diffusion images are blurred due to interpolation. The yellow boxes indicate regions in which the registration is less accurate. In the interest of space, only the right hemisphere is shown.

1. In headings and figures, the different types of microscopy should be differentiated.

Many thanks - we have checked the text to correct this, with many changes highlighted in red throughout the revised manuscript. For example, the caption to Figure 7 has been altered to:

*Figure 7 demonstrates one approach to joint modelling [12] where we combine in-plane orientations from microscopy (here **PLI**) with through-plane information from diffusion MRI (dMRI) to reconstruct 3D hybrid **dMRI-PLI** fibre orientations at the resolution of the microscopy data. For each **PLI** pixel, we compare the **PLI** in-plane orientation to orientations estimated from co-registered diffusion data using the Ball and Stick model [13] which have been projected onto the **PLI** plane (Figure 7a). The **PLI** through-plane angle is then approximated by that from the most similar **BAS** orientation. This produces a hybrid **dMRI-PLI** orientation that is both 3D and at the resolution of the microscopy data.*

2. At the top of p.8, the descriptions of the SLF II need citations to prior work to cement the definition of the tract and the notion of the false positive.

Many thanks for pointing out this omission. The three branches of superior longitudinal fasciculus (SLF I, II, III) based on axonal tracing data in the macaque have been previously described in detail by Schmahmann and Pandya [14], where they also do not observe the “noisy” branching pattern observed in Figure 2d for 64 gradient or 0.6 mm data.

This reference has now been added to Section 2.2 *Ultra-high angular resolution diffusion imaging*:

Instead these tracts appears to have a systematic false positive offshoot (yellow arrows) extending to the superior cortex, which may be indicative of streamlines crossing to the SLF I [14].

2. On p. 10, this claim “Furthermore, in the deep white matter we see different tissue ‘textures’ as well as large scale, ‘wave-like’ undulations of fibres in the corpus callosum (D). Though this a region which is often considered coherently ordered with little fibre dispersion, these data corroborate previous observations of fibre dispersion or incoherence in the corpus callosum [4]” is very interesting, but I dont know what it means. Maybe another figure on these waves would be helpful? I cant tell what feature of the CC is being referenced.

Apologies, the composite figure was indeed too small to fully appreciate the described undulations in the corpus callosum. An additional figure showing the region of interest in high resolution has now been included in Supplementary Figure 4 (see below).

This Figure is also explicitly referenced in the text:

*Furthermore, in the deep white matter we see different tissue ‘textures’ as well as large scale, ‘wave-like’ undulations of fibres in the corpus callosum (D) (see **Supplementary Figure 4** for an enlarged image).*

3. The images of the Gallyas silver match what my concern has always been about using this stain to estimate fiber orientations and build connectomes. Where fibers are relatively sparse, it is possible to distinguish single axons. But where there are many fibers together, the fibers all blur together in a mass, and all sense of individual axons is lost. Does this sound right to the authors? If so, I think stating some of the weaknesses of this approach (it may not be the be-all, end-all) would be helpful.

As the reviewer correctly points out, the myelin stain in the white matter is dense precluding identification of single fibres. However, even within this dense staining, it is possible to distinguish by eye general fibre directions based on the relative stain density, as the stain density is lower but not zero in-between fibres at the surface (see Supplementary Figure 4 above for example). Consequently, using structure tensor analysis - which estimates fibre orientations based on the image gradient over a local neighbourhood - we are able to estimate fibre orientations in the dense white matter. The output orientations follow our neuroanatomical expectations and are highly consistent with those from adjacent PLI, giving confidence in the Gallyas-derived

Supplementary Figure 4: An enlarged version of the undulating fibres in the corpus callosum, as described in Figure 4.

fibre orientations (Manuscript Figure 4).

The following text has been added to Section 2.3.2 *Myelin stained histology* where we compare the outputs from structure tensor analysis to adjacent PLI:

The myelin-stained slides were analysed using structure tensor analysis [15, 16, 17, 18] to estimate the primary fibre orientation per microscopy pixel (Figure 4b)... This image is then compared to an adjacent section imaged with PLI (Figure 4d). Despite the very different manner by which the orientation estimates were derived, the two methods provide corroborating information in both the white matter, where the myelin stain is very dense precluding the identification of individual fibres, and the cortex, where myelinated fibres are less dense.

One of the primary weaknesses of structure tensor analysis is that the user has to select the size of the Gaussian smoothing kernel. Here we utilise a kernel with a standard deviation of 10 pixels (equivalent to $\sim 2.8 \mu\text{m}$). However, future work could consider the impact of different kernel sizes on the structure tensor outputs.

The following text has been added to Section 6.4.2. *Gallyas Silver Staining*, where the structure tensor analysis is described:

One primary limitation of structure tensor analysis is that it requires the user to specify a Gaussian smoothing kernel over which the intensity gradients are calculated. This study utilised a Gaussian kernel with sigma equivalent to 10 pixels, i.e. $\sim 2.8 \mu\text{m}$. Future work could consider the impact of kernels of different sizes.

Reviewer 3

Howard et al. present BigMac, a openly shared multimodal dataset of the macaque brain. This dataset combines in vivo MRI, ex vivo MRI and multiple histological stains. The focus of the authors work here is cross-comparison of diffusion-derived measures of brain anatomy. The high resolution diffusion data, coupled with PLI and fibre orientation estimation from sections stained for myelin is fantastic resource to better understand cerebral fibre organisation and the microstructural underpinnings of DTI. The open sharing of this dataset is commendable will ensure maximum value from this dense resource. While the authors state that the presented dataset will be enriched with additional histological staining, the current data are sufficient to be a unique and valuable addition to multiscale neuroscience.

The MRI diffusion imaging, histological acquisition and preprocessing appear carefully thought out. The authors are clear with in describing the potential artefacts including orbitofrontal lesions, bubbles and mounting-related biases that were addressed during acquisition. The manuscript is primarily a description of the resource, describing the acquisition, highlighting interesting features from visual inspection of the data and some preliminary analyses including comparison of connectivity between 64-direction and 1000-direction diffusion images and comparison of diffusion-microstructural estimates between MRI and histology.

Thank you for your very kind and considered overview of our work.

My main comments relate to these statistical analyses. While I acknowledge they are intended as demonstrations of data quality and the unique analyses afforded by Big Mac, I think these should be strengthened.

1. Figure 2. Comparison of 64 vs 1000 direction DTI. The authors find a 50-150 fold increase in connectivity between lateral homotopic regions. This appears to be an extremely large change, with 3 areas exhibiting orders of magnitude greater numbers of fibres and needs some additional context. Are fibres for these 3 regions expected to be underestimated by 64 direction scans (i.e. some previously recognised issue in DTI)? Do these new values reflect something closer to ground truth using 1000 directions? Is there a way the authors could propose a mechanism as to why these measures are so discordant? If possible, it would also

be powerful to demonstrate the number of connections at 1000 are closer to some external ground truth estimate (e.g. tracer based estimates of connectivity strength)? While I appreciate that streamline length relationship could be playing a role but the huge jump in streamlines shown in 2c panel 5 is beyond that predicted with 2c panel 6, which shows a steadily length-dependent recovery in streamlines.

As you highlight, it can be challenging to ascertain whether streamlines from diffusion MRI represent “true” structural connectivity without additional ex vivo measurements such as tracer data. In the absence of ground truth connectivity estimates within the BigMac brain, Figure 2c highlights the increased streamline density between homotopic regions as functional co-activation of homotopic regions in fMRI data (where previous work demonstrate how pairs of homotopic regions show the strongest functional connections in the brain [19, 20, 21]) suggests these regions likely have underlying structural connectivity. To better understand the large increase in the number of streamlines reaching VACv, TCc and TCi, Supplementary Figures 1-3 (see below) now show the streamline density outputs for data with both 64 and 1000 gradient directions. Here, streamlines were seeded from either the left/right ROI and terminated in the contralateral (right/left) ROI, and the two outputs combined. The left hand images show voxels with $> 1\%$ of all streamlines, where highly non-reproducible and thus noisy streamlines ($< 1\%$ streamline density) are excluded. The right hand images visualise the tract core where only voxels with $> 10\%$ of the total streamlines have been included.

In both the VACv and TCc (Supplementary Figures 1 and 2), we observe how the 1000 gradient data support a large number of streamlines to track through a robustly defined tract core crossing the corpus callosum (pink arrows). The 64 direction data support a pathway through the anterior commissure (orange arrows) that of low probability due to the convoluted, longer route that the streamlines take (more likely a false positive). In contrast, the 1000 direction data supports the more robust, and anatomically more plausible direct route through the corpus callosum. Though the 64 gradient data does also support some streamlines crossing the corpus callosum, the presence of additional secondary/tertiary fibre populations and increased precision of orientation estimates in the 1000 gradient data (Manuscript Figure 2b) facilitates successful tracking through a larger extend of the corpus callosum (with more streamlines able to track through crossing fibre regions either side of the corpus callosum, crossing anterior-posterior pathways to enter the corpus callosum), and a larger number of fibres taking this more plausible route.

In the TCi results (Supplementary Figure 3), we again find that the 1000 gradient data provides a more robust tract core, here through the anterior commissure, as also described in the 64 gradient data. However, at the lower threshold (Supplementary Figure 3 left) the 1000 gradient data also demonstrates fibres tracking through the corpus callosum. Previous work describes i) how severing the corpus callosum greatly reduces interhemispheric functional connectivity [22, 23], providing evidence for a dominant pathway of communication between pairs of homotopic regions through the corpus callosum, and ii) how this effect was greatly mitigated if the anterior commissure was left in tact[22], suggesting a secondary, less dominant pathway of communication through this route. Though these two pathways (through the CC and AC) are observed in our data, the very low number of streamlines recovered for the TCi (only 5 streamlines recovered from the 64 gradient data and 353 streamlines from 1000 gradient data) means our results are noisy and should not be considered at face value. For comparison, 3486 / 196693 streamlines were recovered for the TCc and 279 / 33651 streamlines for the VACv (64 / 1000 gradient data).

Note, the streamline counts in Supplementary Figures 1-3 are similar to, but not the same as, those presented in Figure 2c of the original paper due to the probabilistic nature of the tractography which was rerun for tract visualisation (network analysis only outputs streamline counts for each pair of ROIs). In both cases, the streamline counts show a > 50 increase in the number of streamlines reaching the TCc, TCi and VACv.

Overall, we would argue that the large increase in streamlines taking a more direct route between the homotopic regions of interest via the corpus callosum suggests the 1000 gradient data is likely to be a better representation of the true structural connectivity in the VACv and TCc, whilst the less robust results from the TCi are less conclusive. Future work should consider validating these results against ex vivo “ground” truth measurements from complementary tracer data, which is unfortunately beyond the scope of the current

Supplementary Figure 1: Tractography reconstruction of streamlines connecting the left/right anterior visual area (VACv in the RM parcellation, shown in green) from postmortem diffusion MRI ($b = 10 \text{ ms}/\mu\text{m}^2$, 1 mm isotropic) with either 64 or 1000 gradient directions. We observe how the 1000 gradient data supports a large number of streamlines tracking through the corpus callosum (pink arrows), where secondary/tertiary fibre populations in the 1000 gradient data (which are either absent or poorly estimated from the 64 gradient data) facilitate more robust tracking through crossing fibre regions either side of the callosum. In comparison, the 64 direction data includes a high density of streamlines tracking through the anterior commissure (orange arrows), a more convoluted route that may represent a less dominant or false positive connection.

study.

These new Figures are provided as Supplementary Figures 1-3 in the revised document, and are directly referenced in Section 2.2 *Ultra-high angular resolution diffusion imaging*:

Those with a > 50-fold increase include the ventral part of the anterior visual area (VACv), the inferior temporal cortex (TCi) and the central temporal cortex (TCc), (see Supplementary Figures 1-3 for a more detailed discussion of these results).

2. Figure 6. The plots comparing MRI and histological estimates (fig 6c) would be strengthened by more data points from other sections and some extra statistical analysis. It seems odd given the richness of the dataset that you are restricting these analyses to a few voxels from a single section. Why not increase the sample size for these analyses to include samples from multiple aligned sections across the brain?

Many thanks for this suggestion. We have now repeated the analysis across 20 consecutive PLI slides and 20 histology slides in the right hemisphere. This analysis now includes 3728 voxel-wise comparisons of ODI from postmortem MRI and microscopy, with results now presented as density scatter plots (yellow = high density). Reassuringly, including many voxels did not substantially alter our conclusions.

Supplementary Figure 2: Tractography reconstruction of streamlines connecting the left/right central temporal cortex (TCc in the RM parcellation, shown in green) from postmortem diffusion MRI ($b = 10 \text{ ms}/\mu\text{m}^2$, 1 mm isotropic) with either 64 or 1000 gradient directions. We observe similar results to the VACv, with streamlines mostly tracking through the corpus callosum (pink arrows) in the 1000 gradient data, and a secondary pathway through the anterior commissure (orange arrows) in the 64 gradient data.

Supplementary Figure 3: Tractography reconstruction of streamlines connecting the left/right inferior temporal cortex (TCi in the RM parcellation, shown in green) from postmortem diffusion MRI ($b = 10 \text{ ms}/\mu\text{m}^2$, 1 mm isotropic) with either 64 or 1000 gradient directions. Both the 64 and 1000 gradient data support a tract core crossing the anterior commissure (orange arrows). The 1000 gradient data also supports some streamlines crossing the callosum (pink arrow). These results are less robust due to the relatively low number of streamlines successfully tracking between the left/right regions of interest (< 10 valid streamlines in the 64 gradient data, and a few hundred in the 1000 gradient data), compared to tens of thousands of streamlines in the 1000 gradient direction TCc and VACv results.

Figure 6 has been updated (see below), and these results are described in Results Section 2.5 *Comparing fibre orientation distributions from microscopy and MRI* :

The fibre dispersion can then be quantified using the orientation dispersion index (ODI), which ranges from 0 for perfectly aligned fibres, to 1 for isotropic dispersion [24]. Figure 6c compares dispersion estimates across many white matter voxels (top, covering 20 consecutive PLI and histology slides each), and a subset of voxels from the centrum semiovale (bottom). We see fair correspondence between the dispersion from myelin-stained histology and the diffusion model. Estimates of dispersion from PLI appear less reliable, in line with previous reports [25] and observations in 6b. This may be related to PLI estimating a single orientation per PLI pixel which likely includes many axons (axon diameter is typically $\sim 1 \mu\text{m}$ [26], whilst each PLI pixel with an in-plane resolution of $\sim 4 \mu\text{m}$ and a slice thickness of $50 \mu\text{m}$ covers a volume of $\sim 800 \mu\text{m}^3$). In comparison, the histology data has an order of magnitude higher in-plane resolution ($0.28 \mu\text{m}$, covering a volume of $\sim 4 \mu\text{m}^3$) which may lead to a more faithful estimate of fibre dispersion.

As well as comparing MRI-microscopy equivalents, microscopy can be used to understand indirect relationships with MR parameters. For example, histology dispersion is shown to have a clear negative correlation with fractional anisotropy (FA) from DTI (Figure 6d, [27]). In comparison, dispersion has a weak but significant ($p = 9 \times 10^{-14}$) correlation with microscopic FA (μFA) [28, 29] in the white matter (top) and no significant correlation in the centrum semiovale (bottom), a known deep white matter region of complex dispersion. This is reassuring as the μFA parameter is explicitly meant to be independent of the fibre orientation distribution: in the centrum semiovale, μFA is independent of dispersion, where the small negative correlation across all of white matter may be driven by partial volume effects. In future work when additional microscopy contrasts are added to BigMac, multivariate regressions can be performed to better understand how complex tissue microstructure relates to sensitive but not specific diffusion metrics such as those from the diffusion tensor or other signal models [30].

3. It appears as though some of these results are heavily driven by individual outliers e.g. BAR ODI vs PLI ODI, where a single data point seems to be strongly influencing this line of best fit. Multiple sections would help the impact of one noisy voxel, and statistical significance testing and confidence intervals on the fitted relationships would also be informative.

As suggested, the analysis has now been repeated for many more voxels, and 95% confidence intervals for the line of best fit are also included. A significant linear relationship was found for all plots ($p < 0.05$) except for the bottom right plot comparing histology ODI and microFA (linear regression model subsequently not plotted). The p-value was calculated using an F-test comparing the regression model to a degenerate model consisting of only a constant term.

The following text has been added to Section 2.5:

The regression models in Figure 6c,d reached significance (estimated $p < 0.001$ where $p < 0.05$ is considered significant) for all plots except for 6d bottom left comparing histology ODI with μFA ($p = 0.23$, hence regression model is not shown).

to the Figure 6 caption in the manuscript:

The black lines show the line of best fit, the grey lines indicate the 95% confidence intervals, and r is the correlation coefficient.

and to the Methods Section 6.6 *Comparing fibre orientation distributions from microscopy and MRI*

For each scatter plot, the correlation coefficient r was calculated using MATLABs [31] `fitlm` function, and the p-value calculated using an F-test comparing the regression model to a degenerate model with only a constant term. We performed 3728 voxelwise comparisons in the white matter postmortem, and 2915 voxelwise comparisons in vivo. The centrum semiovale mask contained 78 voxels.

a) Co-registered MRI and microscopy data

b) Comparing fibre orientation distributions on the 2D microscopy plane

c) Correlating MRI and microscopy: direct relationships

d) Indirect relationships

Manscript Figure 6: Comparing information from co-registered **postmortem diffusion MRI** and microscopy. a) Co-registration facilitates qualitative comparisons of microscopy and MRI (here DTI data) when both are warped to a common space. We see clear correspondence between the PLI orientations and the primary eigenvector of the diffusion tensor that has been projected onto the microscopy plane. b) A 2D fibre orientation distribution is extracted on a voxelwise basis from both PLI, **myelin-stained** histology, and the Ball and Rackets model (BAR) for diffusion MRI ($b = 10 \text{ ms}/\mu\text{m}^2$, 1 mm). c) The orientation dispersion index (ODI) from microscopy is correlated with various diffusion metrics: the ODI from the Ball and Rackets model, fractional anisotropy (FA) from the diffusion tensor model (**both calculated from $b = 10 \text{ ms}/\mu\text{m}^2$, 1 mm data**), and μFA from data with multiple tensor encodings ($b = 4, 7 \& 10 \text{ ms}/\mu\text{m}^2$, 1 mm). The top row shows data points **from a white matter mask with ~ 3700 voxels (mask shown in Supplementary Figure 6)**. The bottom row shows only a subset of voxels from a subset of voxels in the centrum semiovale. The black lines show the line of best fit, **the grey lines indicate the 95% confidence intervals**, and r is the correlation coefficient.

4. Could it be made clearer which diffusion MRI data is being compared throughout the manuscript and particularly in this analysis? I think it is all from the post mortem data, but as the authors collected in vivo also it would be useful to clarify which acquisition used (e.g. in the fig 6 caption).

Many thanks for pointing this omission out. Most analyses in the current manuscript were carried out on the postmortem rather than in vivo MRI data, as the postmortem data was acquired specifically for this study. The in vivo data was acquired separately, but is now also made available to add to the richness of the available data.

Clarifications as to which data is analysed for each figure have been made throughout the text, particularly in respect to Figures 5, 6 and 7 which were most ambiguous.

5. Is there a reason why both ex vivo and in vivo DTI weren't included in the fig 6 analyses? This would show the full range of scales and modalities in the resource

It is a little tricky to add the in vivo data directly to Figure 6, as the in vivo and postmortem data do not have directly overlapping voxels, meaning that including all four modalities (in vivo MRI, postmortem MRI, histology and PLI) would require interpolating one of the MRI outputs. Consequently we have included the in vivo results in Supplementary Figure 6. We find similar results when comparing either in vivo MRI or postmortem MRI with the microscopy data, though with higher BAR ODI being estimated in vivo. This is not surprising and may be due to the postmortem data analysed having higher b-value (postmortem MRI $b = 10 \text{ ms}/\mu\text{m}^2$, in vivo MRI $b = 1 \text{ ms}/\mu\text{m}^2$) which results in less contribution of the extra-cellular space to the diffusion signal (which is typically less anisotropic and thus may contribute to higher dispersion), and/or postmortem effects.

The following text has been added to Results Section 2.5 *Comparing fibre orientations from microscopy and MRI*, along with Supplementary Figure 6:

Similar relations were found when correlating in vivo diffusion MRI estimates of fibre dispersion and FA with ODI from both PLI and histology (Supplementary Figure 6).

with additional details in Methods Section 6.6 *Comparing fibre orientations from microscopy and MRI*:

The analysis for Figure 6c,d was repeated with in vivo MRI data. Session 1 data ($b = 1 \text{ ms}/\mu\text{m}^2$, 81 unique gradient directions, 1 mm isotropic) were similarly processed using the Ball and Rackets [32] and diffusion tensor models [27], and estimates of BAR ODI and DTI FA were compared to microscopy ODI estimated in the in vivo MR space.

6. It appears as though the PLI data is underestimating ODI relative to BAR and histology (Fig 6c). BAR and histology have a roughly 1:1 relationship, while the PLI ODI estimates are almost always above the $y = x$ line. Do you have an explanation as to why this is the case? Perhaps this pattern might disappear with more extensive data sampling or maybe there is a systematic bias in one of the measures.

Our results on BigMac data suggest that PLI tends to underestimate fibre fanning or dispersion when compared to equivalent estimates from either structure tensor analysis of histology or diffusion MRI (Figure 6). This result mirrors our previous findings in human postmortem samples, where PLI was compared to PLP-stained sections of $6 \mu\text{m}$ thick. We expect this may at least partly be due to the resolution of the PLI data ($4 \mu\text{m}$ / pixel in plane resolution and a section thickness of $50 \mu\text{m}$), where each PLI pixel contains many axons (typical diameter of $\sim 1 \mu\text{m}$ [26]). Simulations of the PLI signal in the case of crossing fibres demonstrate how the analysis will tend to output an orientation similar to the most dominant fibre orientation of the bundle [33]. This may result in PLI under-estimating dispersion, when compared other modalities. This trend of lower dispersion is found fairly consistently across the brain.

The following text has been added to Section 2.5:

Correlating postmortem MRI metrics against co-registered microscopy

Correlating in vivo MRI metrics against co-registered microscopy

Supplementary Figure 6: Estimates of the Ball and Rackets orientation dispersion index (BAR ODI) and DTI fractional anisotropy (FA) from both postmortem (top) and in vivo (bottom) diffusion MRI are correlated against estimates of ODI from both PLI and myelin histology. The postmortem BAR ODI and FA were estimated from $b = 10 \text{ ms}/\mu\text{m}^2$, 1 mm data. The in vivo MRI metrics were both estimated from $b = 1 \text{ ms}/\mu\text{m}^2$ 1 mm data. Similar relationships were found in vivo and postmortem, though with higher estimates of BAR ODI in vivo. Each point represents a single voxel from the mask shown on the right with a “slab” in MRI space representing 20 consecutive PLI and histology slides in the right hemisphere. The blue-yellow colours indicate a low-high density of points.

Figure 1: In some anterior slices, the cerebellum is divided into many smaller tissue sections making registration time consuming and challenging as each part of the tissue will have to be registered separately.

This may be related to PLI estimating a single orientation per PLI pixel which likely includes many axons (axon diameter is typically $\sim 1 \mu\text{m}$ [26], whilst each PLI pixel with an in-plane resolution of $\sim 4 \mu\text{m}$ and a slice thickness of $50 \mu\text{m}$ covers a volume of $\sim 800 \mu\text{m}^3$). In comparison, the histology data has an order of magnitude higher in-plane resolution ($0.28 \mu\text{m}$, covering a volume of $\sim 4 \mu\text{m}^3$) which may lead to a more faithful estimate of crossing fibres and fibre dispersion.

7. The authors have described cerebellar masking as ongoing work. Could you clarify to what extent the cerebellum has been co-registered to the structural MRI and whether researchers can already carry out joint analyses of these brain areas with this release?

The cerebellum is yet to be registered due to the complexities in this work and the delay it would bring to the release of the data. As shown in Figure 1 above, the convoluted structure of the cerebellar cortex in posterior slides results in many separate tissue sections. Unfortunately, there is no way of knowing whether each section has been mounted correctly, as it is very difficult to keep track of which hemisphere each piece relates to (bar using similarities with previous slides) and it is possible that some sections have been flipped and mounted upside down. Consequently, the registration procedure will require attempting to register each individual partial cerebellar section independently, in both the mounted and flipped configurations, and carefully checking their fidelity to the MRI as well as the registration cost.

To facilitate users wanting to research the cerebellum to perform this registration, we have additionally provided example configuration files for the TIRL protocol as well as instructions on how to generate TIRL warpfields for the BigMac data.

The following text has been added to Section 6.5 *Coregistration of MRI and microscopy data*:

The optimised TIRL protocol was found to produce good results across microscopy slides (see Supplementary Figure 5 for example outputs). However, some users may wish to run their own registration (e.g. to register sections of the cerebellum for which transforms are not yet provided), or optimise the registration further for a specific slide or over a small region of interest. Instructions on how to achieve this, alongside example configuration files, and a script to easily assess the accuracy of the registration for any microscopy slide of interest, are provided in the online documentation and tutorials.

8. Thanks for sharing sample data for this review. Its not clear where the code is going to be made available is there a code repository that I've missed?

Apologies this was unclear. All code is made available on the BigMac GitLab page: <https://git.fmrib.ox.ac.uk/amyh/bigmacanalysis>. This is linked to via the online documentation and the URL is now explicitly provided in the manuscript in the new Section 5 *Code availability*:

Code is available via <https://git.fmrib.ox.ac.uk/amyh/bigmacanalysis>. This includes basic MRI-microscopy tutorials as well as scripts related to data preprocessing, or reproducing the analyses presented.

9. I spotted some minor typos / clarifications:

Line 28 Scientific communities ↵ communities

Line 44 consider rewording: These analyses require complex computational signal modelling with many strong assumptions. Do you mean the assumptions are strong, or that the modelling is strongly dependent on assumptions?

Line 57 has co-registered MRI and microscopy data for to facilitate meaningful voxelwise comparisons

383 corss > cross

449 Is 201 degrees air temperature correct? That sounds hot, like a temperature at which the tissue (and researchers) might begin to cook.

Many thanks for catching these. Each have now been addressed.

Konrad Wagstyl

- [1] D. Le Bihan, J. Delannoy, R. L. Levin, Temperature mapping with MR imaging of molecular diffusion: application to hyperthermia, *Radiology* doi:10.1148/radiology.171.3.2717764.
- [2] I. N. Huszar, M. Pallebage-Gamarallage, S. Bangerter-Christensen, H. Brooks, S. Fitzgibbon, S. Foxley, M. Hiemstra, A. F. Howard, S. Jbabdi, D. Z. Kor, A. Leonte, J. Mollink, A. Smart, B. C. Tendler, M. R. Turner, O. Ansoerge, K. L. Miller, M. Jenkinson, Tensor image registration library: Deformable registration of standalone histology images to wholebrain postmortem MRI data, *NeuroImage* 265 (2023) 119792. doi:10.1016/J.NEUROIMAGE.2022.119792.
- [3] M. P. Heinrich, M. Jenkinson, M. Bhushan, T. Matin, F. V. Gleeson, S. M. Brady, J. A. Schnabel, MIND: Modality independent neighbourhood descriptor for multi-modal deformable registration, *Medical Image Analysis* 16 (7) (2012) 1423–1435. doi:10.1016/j.media.2012.05.008.
- [4] M. Menzel, K. Michielsen, H. De Raedt, J. Reckfort, K. Amunts, M. Axer, A Jones matrix formalism for simulating three-dimensional polarized light imaging of brain tissue, *Journal of the Royal Society Interface* 12 (111) (2015) 20150734. doi:10.1098/rsif.2015.0734.
- [5] M. Koike-Tani, T. Tani, S. B. Mehta, A. Verma, R. Oldenbourg, Polarized light microscopy in reproductive and developmental biology, *Molecular Reproduction and Development* 82 (7-8) (2015) 548–562. doi:10.1002/MRD.22221.
- [6] X. R. Huang, R. W. Knighton, Microtubules Contribute to the Birefringence of the Retinal Nerve Fiber Layer, *Investigative Ophthalmology & Visual Science* 46 (12) (2005) 4588–4593. doi:10.1167/IOVS.05-0532.
- [7] F. Matuschke, K. Amunts, M. Axer, fastPLI: A Fiber Architecture Simulation Toolbox for 3D-PLI, *Journal of Open Source Software* 6 (61) (2021) 3042. doi:10.21105/JOSS.03042.
- [8] F. O. Schmitt, R. S. Bear, The Ultrastructure of the Nerve Myelin Sheath, *Biological Reviews* 14 (1) (1939) 27–50. doi:10.1111/J.1469-185X.1939.TB00922.X.
- [9] M. Koike-Tani, T. Tominaga, R. Oldenbourg, T. Tani, Birefringence Changes of Dendrites in Mouse Hippocampal Slices Revealed with Polarizing Microscopy, *Biophysical Journal* 118 (10) (2020) 2366–2384. doi:10.1016/J.BPJ.2020.03.016.
- [10] K. Amunts, C. Lepage, L. Borgeat, H. Mohlberg, T. Dickscheid, M. É. Rousseau, S. Bludau, P. L. Bazin, L. B. Lewis, A. M. Oros-Peusquens, N. J. Shah, T. Lippert, K. Zilles, A. C. Evans, BigBrain: An ultrahigh-resolution 3D human brain model, *Science* 340 (6139) (2013) 1472–1475. doi:10.1126/science.1235381.
- [11] B. C. Tendler, T. Hanayik, O. Ansoerge, S. Bangerter-Christensen, G. S. Berns, M. F. Bertelsen, K. L. Bryant, S. Foxley, P. Martijn van den Heuvel, A. F. Howard, I. N. Huszar, A. A. Khrapitchev, A. Leonte, P. R. Manger, R. A. Menke,

- J. Mollink, D. Mortimer, M. Pallebage-Gamarallage, L. Roumazeilles, J. Sallet, L. H. Scholtens, C. Scott, A. Smart, M. R. Turner, C. Wang, S. Jbabdi, R. B. Mars, K. L. Miller, The Digital Brain Bank, an open access platform for post-mortem datasets, *eLife* 11. doi:10.7554/ELIFE.73153.
- [12] A. Howard, S. Jbabdi, K. AA, J. Sallet, G. Daubney, J. Mollink, C. Scott, N. Sibson, K. Miller, The BigMac dataset: ultra-high angular resolution diffusion imaging and multi-contrast microscopy of a whole macaque brain, ISMRM 27th Annual Meeting.
- [13] T. E. J. Behrens, M. W. Woolrich, M. Jenkinson, H. Johansen-Berg, R. G. Nunes, S. Clare, P. M. Matthews, J. M. Brady, S. M. Smith, Characterization and Propagation of Uncertainty in Diffusion-Weighted MR Imaging, *Magnetic Resonance in Medicine* 50 (5) (2003) 1077–1088. doi:10.1002/mrm.10609.
- [14] J. D. Schmahmann, D. N. Pandya, *Fiber Pathways of the Brain, Fiber Pathways of the Brain* (2006) 1–654doi:10.1093/ACPROF:OSO/9780195104233.001.0001. URL <https://academic.oup.com/book/362>
- [15] J. Bigun, T. Bigun, K. Nilsson, Recognition by symmetry derivatives and the generalized structure tensor, *IEEE Transactions on Pattern Analysis and Machine Intelligence* 26 (12) (2004) 1590–1605. doi:10.1109/tpami.2004.126.
- [16] M. D. Budde, J. A. Frank, Examining brain microstructure using structure tensor analysis of histological sections, *NeuroImage* 63 (1) (2012) 1–10. doi:10.1016/j.neuroimage.2012.06.042.
- [17] M. D. Budde, J. Annese, Quantification of anisotropy and fiber orientation in human brain histological sections, *Frontiers in Integrative Neuroscience* 7 (2013) 3. doi:10.3389/fnint.2013.00003.
- [18] A. Seehaus, A. Roebroek, M. Bastiani, L. Fonseca, H. Bratzke, N. Lori, A. Vilanova, R. Goebel, R. Galuske, Histological validation of high-resolution DTI in human post mortem tissue, *Frontiers in Neuroanatomy* 9 (2015) 1–12. doi:10.3389/fnana.2015.00098.
- [19] J. Mollink, S. M. Smith, L. T. Elliott, M. Kleinnijenhuis, M. Hiemstra, F. Alfaro-Almagro, J. Marchini, A. M. van Cappellen van Walsum, S. Jbabdi, K. L. Miller, The spatial correspondence and genetic influence of interhemispheric connectivity with white matter microstructure, *Nature Neuroscience* 2019 22:5 22 (5) (2019) 809–819. doi:10.1038/s41593-019-0379-2.
- [20] K. Shen, B. Mišić, B. N. Cipollinic, G. Bezgin, M. Buschkuehl, R. M. Hutchison, S. M. Jaeggi, E. Kross, S. J. Peltier, S. Everling, J. Jonides, A. R. McIntosh, M. G. Berman, Stable long-range interhemispheric coordination is supported by direct anatomical projections, *Proceedings of the National Academy of Sciences of the United States of America* 112 (20) (2015) 6473–6478. doi:10.1073/PNAS.1503436112.
- [21] D. E. Stark, D. S. Margulies, Z. E. Shehzad, P. Reiss, A. M. Kelly, L. Q. Uddin, D. G. Gee, A. K. Roy, M. T. Banich, F. X. Castellanos, M. P. Milham, Regional Variation in Interhemispheric Coordination of Intrinsic Hemodynamic Fluctuations, *Journal of Neuroscience* 28 (51) (2008) 13754–13764. doi:10.1523/JNEUROSCI.4544-08.2008.
- [22] J. X. O’Reilly, P. L. Croxson, S. Jbabdi, J. Sallet, M. P. Noonan, R. B. Mars, P. G. Browning, C. R. Wilson, A. S. Mitchell, K. L. Miller, M. F. Rushworth, M. G. Baxter, Causal effect of disconnection lesions on interhemispheric functional connectivity in rhesus monkeys, *Proceedings of the National Academy of Sciences of the United States of America* 110 (34) (2013) 13982–13987. doi:10.1073/PNAS.1305062110.
- [23] J. L. Roland, A. Z. Snyder, C. D. Hacker, A. Mitra, J. S. Shimony, D. D. Limbrick, M. E. Raichle, M. D. Smyth, E. C. Leuthardt, On the role of the corpus callosum in interhemispheric functional connectivity in humans, *Proceedings of the National Academy of Sciences of the United States of America* 114 (50) (2017) 13278–13283. doi:10.1073/PNAS.1707050114.
- [24] H. Zhang, T. Schneider, C. A. Wheeler-Kingshott, D. C. Alexander, NODDI: Practical in vivo neurite orientation dispersion and density imaging of the human brain, *NeuroImage* 61 (4) (2012) 1000–1016. doi:10.1016/j.neuroimage.2012.03.072.
- [25] J. Mollink, M. Kleinnijenhuis, A. M. van Cappellen van Walsum, S. N. Sotiropoulos, M. Cottaar, C. Mirfin, M. P. Heinrich, M. Jenkinson, M. Pallebage-Gamarallage, O. Ansorge, S. Jbabdi, K. L. Miller, Evaluating fibre orientation dispersion in white matter: Comparison of diffusion MRI, histology and polarized light imaging, *NeuroImage* 157 (2017) 561–574. doi:10.1016/j.neuroimage.2017.06.001.
- [26] J. Veraart, D. Nunes, U. Rudrapatna, E. Fieremans, D. K. Jones, D. S. Novikov, N. Shemesh, Noninvasive quantification of axon radii using diffusion MRI, *eLife* 9 (1). doi:10.7554/eLife.49855.
- [27] P. J. Basser, J. Mattiello, D. LeBihan, MR diffusion tensor spectroscopy and imaging, *Biophysical Journal* 66 (1) (1994) 259–267. doi:10.1016/S0006-3495(94)80775-1.
- [28] S. Lasič, F. Szczepankiewicz, S. Eriksson, M. Nilsson, D. Topgaard, Microanisotropy imaging: quantification of microscopic diffusion anisotropy and orientational order parameter by diffusion MRI with magic-angle spinning of the q-vector, *Frontiers in Physics* 2 (2014) 11. doi:10.3389/fphy.2014.00011.
- [29] F. Szczepankiewicz, S. Lasič, D. van Westen, P. C. Sundgren, E. Englund, C.-F. Westin, F. Ståhlberg, J. Lätt, D. Topgaard, M. Nilsson, Quantification of microscopic diffusion anisotropy disentangles effects of orientation dispersion from microstructure: Applications in healthy volunteers and in brain tumors, *NeuroImage* 104 (2015) 241–252. doi:10.1016/j.neuroimage.2014.09.057.
- [30] D. Z. Kor, S. Jbabdi, I. N. Huszar, J. Mollink, B. C. Tandler, S. Foxley, C. Wang, C. Scott, A. Smart, O. Ansorge, M. Pallebage-Gamarallage, K. L. Miller, A. F. Howard, An automated pipeline for extracting quantitative histological metrics for voxelwise MRI-histology comparisons, *bioRxiv*doi:10.1101/2022.02.10.479718.
- [31] T. MathWorks, MATLAB (R2017b) (2017). doi:10.1007/s10766-008-0082-5.
- [32] S. N. Sotiropoulos, T. E. Behrens, S. Jbabdi, Ball and rackets: Inferring fiber fanning from diffusion-weighted MRI, *NeuroImage* 60 (2) (2012) 1412–1425. doi:10.1016/j.neuroimage.2012.01.056.
- [33] M. Dohmen, M. Menzel, H. Wiese, J. Reckfort, F. Hanke, U. Pietrzyk, K. Zilles, K. Amunts, M. Axer, Understanding fiber mixture by simulation in 3D Polarized Light Imaging, *NeuroImage* 111 (2015) 464–475. doi:10.1016/J.NEUROIMAGE.2015.02.020.

Reviewer #1 (Remarks to the Author):

The authors have completely addressed all of my questions and I believe that the updated manuscript is ready for publication.

Reviewer #2 (Remarks to the Author):

The authors have done a nice job responding to all reviewer comments.

Reviewer #3 (Remarks to the Author):

Thanks to the the authors for responding to all comments thoughtfully and thoroughly. I have no further suggested changes.

We thank all the reviewers for their consideration of our manuscript. No further changes have been made.

Reviewer 1 (Remarks to the Author):

The authors have completely addressed all of my questions and I believe that the updated manuscript is ready for publication.

Thank you.

Reviewer 2 (Remarks to the Author):

The authors have done a nice job responding to all reviewer comments.

Thank you.

Reviewer 3 (Remarks to the Author):

Thanks to the the authors for responding to all comments thoughtfully and thoroughly. I have no further suggested changes.

Thank you.